# A Framework for Finding Local Saddle Points in Two-Player Zero-Sum Black-Box Games

## Abstract

Saddle point optimization is a critical problem employed in numerous real-world applications, including portfolio optimization, generative adversarial networks, and robotics. It has been extensively studied in cases where the objective function is known and differentiable. Existing work in *black-box* settings with unknown objectives that can only be sampled either assumes convexity-concavity in the objective to simplify the problem or operates with noisy gradient estimators. In contrast, we introduce a framework inspired by Bayesian optimization which utilizes Gaussian processes to model the unknown (potentially nonconvex-nonconcave) objective and requires only zeroth-order samples. Our approach frames the saddle point optimization problem as a two-level process which can flexibly integrate existing and novel approaches to this problem. The upper level of our framework produces a model of the objective function by sampling in promising locations, and the lower level of our framework uses the existing model to frame and solve a general-sum game to identify locations to sample. This lower level procedure can be designed in complementary ways, and we demonstrate the flexibility of our approach by introducing variants which appropriately trade off between factors like runtime, the cost of function evaluations, and the number of available initial samples. We experimentally demonstrate these algorithms on synthetic and realistic datasets, showcasing their ability to efficiently locate local saddle points in these contexts.

## 1  Introduction

We consider the problem of finding saddle points for smooth two-player zero-sum games of the form

$$\text{PLAYER 1:} \ \min_x f(x,y) \qquad \text{PLAYER 2:} \ \min_y -f(x,y) \qquad x \in \mathbb{R}^{n_x}, y \in \mathbb{R}^{n_y} \qquad (1)$$

with an unknown, nonconvex-nonconcave objective $f$. We assume that we can draw noisy zeroth-order samples of $f$ via a possibly expensive process given coordinates $(x,y)$, where $x \in \mathbb{R}^{n_x}, y \in \mathbb{R}^{n_y}$.

Saddle points are points at which the function $f$ is *simultaneously* a minimum along the $x$-coordinate and a maximum along the $y$-coordinate. Such points specialize the well-known Nash equilibrium concept to the setting of two-player, zero-sum games. Saddle point optimization (Tind, 2009) is widely used in real-world applications like economics (Luxenberg et al., 2022), machine learning (Goodfellow et al., 2020), robotics (Agarwal et al., 2023), communications (Moura & Hutchison, 2019), chemistry (Henkelman et al., 2000), and more.

Zero-sum games have been widely studied for known and differentiable objective functions. However, this assumption does not encompass numerous real-world situations with nonconvex-nonconcave objectives which may be unknown and can only be sampled. Such objectives are often referred to as "black-box." For example, in robust portfolio optimization, the goal is to create portfolios resistant to stock market fluctuations (Nyikosa, 2018), which are inherently random and difficult to model but can be sampled in a black-box fashion through trial and error. Similar problems arise in various physical settings, such as robotics (Lizotte et al., 2007) and communication networks (Qureshi & Khan, 2023). Motivated by these real-world examples in nonconvex-nonconcave black-box settings, we present a flexible framework that seeks to identify a saddle point, $(x^*, y^*)$, such that $f(x^*, y) \leq f(x^*, y^*) \leq f(x, y^*)$, for all $x, y$ in its neighborhood.

Most previous research in this area has focused on solving minimax problems (Bogunovic et al., 2018; Fröhlich et al., 2020; Wang et al., 2022), which take the form $\min_x \max_y f(x,y)$. The difference

between minimax and saddle points is subtle: a minimax point achieves the best worst-case outcome for the minimizer (i.e., a Stackelberg equilibrium). In contrast, at a saddle point, the best worst-case and worst best-case outcomes coincide (i.e., a Nash equilibrium). Solutions to minimax problems in general nonconvex-nonconcave settings are not necessarily Nash, and encode an leader-follower hierarchy which is not present for the saddle point concept. In settings like racing, chess, and resource allocation, where rational, adversarial actors make decisions simultaneously, equilibria are best described as saddle points.

Many previous works in saddle point optimization assume convex-concave objectives (v. Neumann, 1928; Korpelevich, 1976; Tseng, 1995; Nemirovski, 2004), for which every minimax point is a saddle and vice versa because the best worst-case and worst best-case always coincide. However, this equivalence does not hold in general nonconvex-nonconcave settings. Notably, some prior works addressing black-box convex-concave settings use zeroth-order samples (Maheshwari et al., 2022). Lastly, finding global saddle points remains an open problem in general settings, so our work specifically focuses on discovering local saddle points, as detailed in Remark 3.5.

In contrast to previous works, we approach this problem in the spirit of bilevel Bayesian optimization: at a high-level, we use Gaussian processes to build a surrogate model for the black-box function $f(x, y)$ by sampling points $(x, y)$ at promising locations, and at a low-level, we identify these sample points by solving *general-sum* games defined on the surrogate model. Specifically, the low-level game selects these samples by seeking local Nash points (Defn. 3.2) of these two-player general-sum games. The high-level optimizer then aims to ensure that in the limit, these samples converge to local saddle points of the black-box problem. We present our **contributions** as follows.

1. We present the first black-box technique for saddle point optimization on nonconvex-nonconcave objectives based on zeroth-order information. While prior works exist that find saddle points in black-box settings or on nonconvex-nonconcave objectives or with zeroth-order information, our work is the first, to our knowledge, that achieves all three simultaneously.

2. We use our framework to propose a set of algorithms for the lower-level game, with each variant catering to a different real-world case based on uncertainty and sampling cost. Thus, our approach allows for versatility in trading off between factors like ease of sampling, exploration and exploitation, and provides a template for future work in this area.

3. We experimentally demonstrate our algorithms' effectiveness on a variety of challenging synthetic and realistic datasets. As with prior work in black-box optimization, a key limitation of our method is that it is appropriate for only relatively low-dimensional spaces.

## 2 RELATED WORK AND PRELIMINARIES

**Saddle Point Optimization:** Saddle point problems are widely studied in the game theory (Başar & Olsder, 1998; Cherukuri et al., 2017), optimization (Dauphin et al., 2014; Pascanu et al., 2014), and machine learning communities (Benzi et al., 2005; Jin et al., 2021). We note three previous varieties of algorithms in the area of nonconvex-nonconcave saddle point optimization which guarantee convergence to local saddle points rather than stationary points. Of these, Adolphs et al. (2019) and Gupta et al. (2024) introduce algorithms which solve for saddle points in deterministic settings and neither approach handles unknown objectives. Mazumdar et al. (2019) introduces local symplectic surgery (LSS), a method that, when it converges, provably does so to a local saddle point given access to first-order and second-order derivative measurements by the sampler. However, derivative information is not always available in systems of interest. Moreover, while zeroth-order samplers can estimate noisy first-order (and second-order) derivatives via finite differencing, the extensive sampling requirements for such an approach is prohibitive when sampling is expensive. By contrast, our work proposes an extensible zeroth-order framework for black-box saddle point optimization, and we provide extensive experimental evaluation of several variants tailored to distinct settings.

**Gaussian Process (GP):** A GP (Rasmussen, 2003), denoted by $\mathrm{GP}(\mu(\cdot), \Sigma(\cdot, \cdot))$, is a set of random variables, such that any finite sub-collection $\{f(x_i)\}_{i=1}^n$ is jointly Gaussian with mean $\mu(x_i) = \mathbb{E}[f(x_i)]$, and covariance $\Sigma = \mathbb{E}[(f(x_i) - \mu(x_i))(f(x_j) - \mu(x_j))], \forall i, j \in \{1, \ldots, n\}$.

GPs are primarily used for regression tasks, where they predict an underlying function, $f : \mathbb{R}^n \to \mathbb{R}$, given some previously observed noisy measurements. That is, for any inputs $x_1, \ldots, x_n \in \mathcal{X} \subseteq \mathbb{R}^k$, and the corresponding noisy measurements, $r_1, \ldots, r_n \in \mathbb{R}$, the vector $r = [r_1, r_2, \ldots, r_n]^\top$ is modeled as multivariate Gaussian distribution with mean vector $\mu$ (typically assumed to be

zero), and covariance matrix $\Sigma \in \mathbb{R}^{n \times n}$. The covariance matrix, $\Sigma$, is calculated as follows: $\Sigma_{i,j} = K(x_i, x_j), \forall i, j \in \{1, \ldots, n\}$, where $K(\cdot, \cdot)$ is the kernel function. Typically, it is assumed that errors $z_i = r_i - f(x_i)$ are normally, independently, and identically distributed, i.e. $z_i \in \mathcal{N}(0, \sigma_z^2)$. At a given test point $x_*$, we may compute the marginal distribution of $f(x_*)$ given $r$ via

$$f(x_*) \mid r \sim \mathcal{N}(\underbrace{k_*^\top (\Sigma + \sigma_z^2 I)^{-1} r}_{\mu_t(x_*)}, \underbrace{K(x_*, x_*) - k_*^\top (\Sigma + \sigma_z^2 I)^{-1} k_*}_{\sigma_t(x_*)}), \tag{2}$$

where $k_* = [K(x_1, x_*), \cdots, K(x_n, x_*)]^\top$. A more detailed description of (2) can be found in Rasmussen (2003). We note that GP estimates are smooth and thus standard gradient-based algorithms can be deployed on them to estimate solutions to optimization problems. For typical (e.g., squared exponential) kernels, the number of samples required for GP regression increases exponentially in the number of dimensions; thus, GPs are appropriate for only relatively low dimensional spaces.

**Bayesian Optimization (BO) with Gaussian Processes:** Močkus (1975); Brochu et al. (2010b); Shahriari et al. (2016), is a sequential search method for maximizing an unknown objective function $f : \mathbb{R}^k \to \mathbb{R}$ with as few evaluations as possible. It starts with initializing a prior over $f$ and uses an acquisition function to select the next point $x_t$ given the history of observations, $f(x_1), \ldots, f(x_{t-1})$. The unknown objective, $f$, is sampled at $x_t$, and its observed value $f(x_t)$ is used to update the current estimate of $f$. Typically, $f$ is modeled as a GP, and the GP prior is updated with new samples. One common acquisition function, used in the GP-UCB algorithm, is the Upper Confidence Bound $\text{UCB}_t(x) = \mu_t(x) + \beta_t \sigma_t(x)$. GP-UCB (Srinivas et al., 2010) has been used in a variety of settings including robotics (Deisenroth et al., 2013), chemistry (Westermayr & Marquetand, 2021), user modeling (Brochu et al., 2010a), and reinforcement learning (Cheung et al., 2020). A high $\beta_t$ parameter in $\text{UCB}_t$ implies a more optimistic maximizer (i.e. favoring exploration) in the presence of uncertainty. The UCB acquisition function combines the estimated mean, $\mu_t(x)$, and the estimated standard deviation, $\sigma_t(x)$, of the unknown objective function $f$ at point $x$ at iteration $t$. Analogously, we can also define the Lower Confidence Bound ($\text{LCB}_t(x) = \mu_t(x) - \beta_t \sigma_t(x)$).

## 3 PROBLEM FORMULATION

**Problem Setup:** We consider the two-player, zero-sum game in (1), and focus on the case in which the objective $f$ is an unknown function defined on the domain $\mathbb{R}^{n_x} \times \mathbb{R}^{n_y}$, and can only be realized through (possibly expensive, noisy) evaluations. That is, we query the objective at a point $(x, y) \in \mathbb{R}^{n_x} \times \mathbb{R}^{n_y}$ and observe a noisy sample $r = f(x, y) + z$, where $z \sim \mathcal{N}(0, \sigma_z^2)$. Although $f$ itself is unknown, we will assume that it is smooth and can be differentiated twice. Our goal is to find Local Saddle Points (LSPs) of $f$.

Our proposed framework will consist of two stages: at the lower level, we will solve a general-sum game defined on a GP surrogate model to identify (local) Nash points, and at the high level we will sample $f$ at those points and refine the GP surrogate model. To frame this problem formally, we must discuss the relationship between Nash points which solve general-sum games and saddle points which solve zero-sum games. For a two-player general-sum game, where player 1 is minimizing function $f_1$, and player 2 is minimizing function $f_2$, a Nash point, $(x^*, y^*)$, is defined as:

**Definition 3.1** (Global Nash Point (GNP) for Two-Player General-Sum Game)**.** (Başar & Olsder, 1998, Defn. 2.1) Point $(x^*, y^*)$ is a global Nash point of objectives $f_1$, $f_2$ if, for all $x, y \in \mathbb{R}^{n_x} \times \mathbb{R}^{n_y}$,

$$f_1(x^*, y^*) \leq f_1(x, y^*), f_2(x^*, y^*) \leq f_2(x^*, y). \tag{3}$$

At a GNP, variables $x$ and $y$ cannot change their respective values without achieving a less favorable outcome. Finding a GNP is computationally intractable in nonconvex settings (as in global nonconvex optimization), so we seek a Local Nash Point (LNP), where this property need only hold within a small neighborhood. A LNP is characterized by first- and second-order conditions.

**Definition 3.2** (LNP for Two-Player General-Sum Game)**.** (Ratliff et al., 2016, Defn. 1) Let $\| \cdot \|$ denote a vector norm. A point, $(x^*, y^*)$, is a local Nash point of cost functions $f_1$ and $f_2$ if there exists a $\tau > 0$ such that for any $x$ and $y$ satisfying $\|x - x^*\| \leq \tau$ and $\|y - y^*\| \leq \tau$, we have (3).

**Proposition 3.3** (First-order Necessary Condition)**.** *(Ratliff et al., 2016, Prop. 1) For differentiable $f_1$ and $f_2$, a local Nash point $(x^*, y^*)$ satisfies $\nabla_x f_1(x^*, y^*) = 0$ and $\nabla_y f_2(x^*, y^*) = 0$.*

**Proposition 3.4** (Second-order Sufficient Condition)**.** *(Ratliff et al., 2016, Defn. 3) For twice-differentiable $f_1$ and $f_2$, if $(x, y)$ satisfies the conditions in Prop. 3.3, $\nabla_{xx}^2 f_1(x, y) \succ 0$, and $\nabla_{yy}^2 f_2(x, y) \succ 0$, then it is a strict local Nash point.*

*Remark* 3.5 (Nash Point is a Saddle Point when $f_1 = -f_2$). If $f = f_1 = -f_2$, then the point $(x^*, y^*)$ is a global saddle point of $f$ when Defn. 3.1 holds and a local saddle point when Defn. 3.2 holds. A local saddle point is characterized by the same first- and second-order conditions defined in Prop. 3.3 and Prop. 3.4, respectively. Henceforth, we use the term saddle point to refer to Nash points in zero-sum games and refer to Defns. 3.1 and 3.2 and Props. 3.3 and 3.4 for Nash and saddle points.

## 4 BLACK-BOX ALGORITHMS FOR FINDING LOCAL SADDLE POINTS

We summarize our Bayesian Optimization (BO)-inspired bilevel framework for identifying local saddle points in the black-box setting. Let $\mu_t$ and $\Sigma_t$, respectively, define mean and covariance functions that estimate the unknown objective $f$ as a Gaussian process based on a dataset $\mathcal{S}_t = \{(x_i, y_i, r_i)\}_{i=1}^t$ where $r_i \in \mathbb{R}$ is a (potentially noisy) sample of $f$ at point $(x_i, y_i)$. We define a zero-sum game, which we refer to as the *high-level* game,

$$\text{PLAYER 1:} \quad x^* = \arg\min_x \mu_t(x, y) \quad \text{PLAYER 2:} \quad y^* = \arg\min_y -\mu_t(x, y) \qquad (4)$$

This game has two purposes: primarily, it seeks to solve for a LSP of the original problem (1). Doing so requires solving the secondary problem of refining the GP estimate by strategically sampling $f$ to form $\mathcal{S}_{t+1} = \mathcal{S}_t \cup \{x_{t+1}, y_{t+1}, r_{t+1}\}$ at iteration $t + 1$. To identify promising points, we solve a *low-level* general-sum game

$$\text{PLAYER 1:} \quad \bar{x}^* = \arg\min_x \text{LCB}_t(x, y) \quad \text{PLAYER 2:} \quad \bar{y}^* = \arg\min_y -\text{UCB}_t(x, y) \qquad (5)$$

for a local Nash point. As $\mu_t$ and $\Sigma_t$ are smooth functions, we can solve (5) by deploying standard gradient-based algorithms on them to solve the lower-level game for first-order stationary points.

Critically, our method relies on an observation about the relationship between this general-sum LNP and the zero-sum LSP $(x^*, y^*)$ we wish to find. In the limit of infinite samples in the neighborhood of $(x^*, y^*)$, when the GP surrogate converges to $f$, then the uncertainty $\sigma$ converges to zero and $\text{LCB}_t$ and $\text{UCB}_t$ converge to the mean $\mu_t$, which converges to $f$ and leads problems (4) and (5) to coincide. These games optimize $\mu_t$, which is an estimate of $f$, so note that any solutions will be approximate.

### 4.1 DEFINING AND SOLVING THE LOW-LEVEL GAME FOR LOCAL NASH POINTS

Extending the familiar "optimization in the face of uncertainty" principle from BO (Snoek et al., 2012) and active learning (Yang et al., 2015), we construct the low-level game in (5) so that each player minimizes a lower bound on its nominal performance index. As in active learning, this design is intended to encourage "exploration" of promising regions of the optimization landscape early on, before "exploiting" the estimated GP model. These bounds, $\text{LCB}_t$ and $\text{UCB}_t$, are constructed at each iteration $t$ of the high-level game.

Finding LNPs is computationally intractable in general; therefore, in practice we seek only points which satisfy the first-order conditions of Prop. 3.3. To find this solution, we introduce a new function, $G_t^{\text{CB}}(x, y) : \mathbb{R}^{n_x} \times \mathbb{R}^{n_y} \to \mathbb{R}^k$, whose roots coincide with these first-order Nash points. The superscript CB and subscript $t$ signify that we are finding the roots with confidence bounds (CB) at iteration $t$. Specifically, we seek the roots of the following nonlinear system of (algebraic) equations:

$$G_t^{\text{CB}}(x, y) = \begin{bmatrix} \nabla_x \text{LCB}_t(x, y) \\ -\nabla_y \text{UCB}_t(x, y) \end{bmatrix} = 0. \qquad (6)$$

**The LLGAME Algorithm:** In Alg. 1, we present our approach to solving the *low-level* game, which we refer to as LLGAME. LLGAME utilizes the current confidence bounds, $(\text{LCB}_t, \text{UCB}_t)$, to determine the local Nash points by finding roots of $G_t^{\text{CB}}$ using Newton's method. The LLGAME function does not make any new queries of $f$; instead, it uses the most up-to-date confidence bounds to identify the local Nash points. LLGAME takes an initial point, $(x, y)$, and current confidence bounds, $(\text{LCB}_t, \text{UCB}_t)$, as inputs. Starting from line 2, the algorithm iteratively updates the point, $(\bar{x}_t, \bar{y}_t)$, using (7)—discussed below—and halts when a merit function, $M_t^{\text{CB}}(\bar{x}_t, \bar{y}_t)$, and therefore the gradients $\nabla_{\bar{x}_t} \text{LCB}_t(\bar{x}_t, \bar{y}_t)$ and $\nabla_{\bar{y}_t} \text{UCB}_t(\bar{x}_t, \bar{y}_t)$, are sufficiently small. Ultimately in line 4, the function returns the final point, $(\bar{x}^*, \bar{y}^*)$, once it discovers a local Nash point.

**Defining Convergence (line 2):** To gauge the progress towards a root of $G_t^{\text{CB}}(x, y)$, we employ a *merit function*, a scalar-valued function of $(x, y)$, which equals zero at a root and grows unbounded far

**Function**
    LLGAME $((x, y, \text{LCB}, \text{UCB}))$ :
  1:     $\bar{x}_0, \bar{y}_0 = x, y$.
  2:     **while** $M_t^{\text{CB}}(\bar{x}_t, \bar{y}_t) \geq \epsilon$ **do**
  3:        Get next iterate $(\bar{x}_{t+1}, \bar{y}_{t+1})$ using LCB, UCB as shown in (7).
       **end**
  4:     **Return** $\bar{x}^*, \bar{y}^*$.
**End Function**

**Algorithm 1:** LLGAME

**Input:** $\mathcal{S}_0$, GP prior $(\mu_0, \sigma_0)$, $\epsilon$.
  1: Start with initial point
     $(x_0, y_0) = \arg\min_{(x,y) \in \mathcal{S}_0} M_t^{\mu}(x, y)$.
  2: **while** $M_t^{\mu}(x_t, y_t) \geq \epsilon$ **do**
  3:     $(x_{t+1}, y_{t+1}) = \text{LLGAME}(x_t, y_t, \text{LCB}_t, \text{UCB}_t)$.
  4:     Sample $f(x_{t+1}, y_{t+1})$ and add to $\mathcal{S}_{t+1}$.
  5:     Update $\mu_{t+1}, \sigma_{t+1}, \text{LCB}_{t+1}, \text{UCB}_{t+1}$.
  6:     $(x_t, y_t) = (x_{t+1}, y_{t+1})$.
     **end**
  7: **Return** Saddle point $x^*, y^*$.

**Algorithm 2:** Bayesian Saddle Point (BSP) Algorithm

away from a root. Specifically, we use the squared $\ell_2$ norm as the merit function, i.e., $M_t^{\text{CB}}(x, y) = \frac{1}{2}\|G_t^{\text{CB}}(x, y)\|_2^2$; each LNP of (5) is a global minimizer of $M$.

**Nonlinear Root-finding with Newton's Method (line 3):** To find the roots of $G_t^{\text{CB}}(x, y)$, our work employs Newton's method, which is an iterative method that is widely utilized for solving nonlinear systems. The Newton step, $p_t(x, y)$, is obtained by linearly approximating the function, $G_t^{\text{CB}}$, with its Jacobian matrix $J_t(x, y)$ at the current estimate $(x, y)$ and identifying the root of that approximation. The step $p_t(x, y)$ therefore satisfies:

$$J_t(x, y)p_t(x, y) = -G_t^{\text{CB}}(x, y), \quad \text{with } J_t(x, y) = \begin{bmatrix} \nabla_{x,x}^2 \text{LCB}_t & \nabla_{x,y}^2 \text{LCB}_t \\ -\nabla_{y,x}^2 \text{UCB}_t & -\nabla_{y,y}^2 \text{UCB}_t \end{bmatrix}. \quad (7)$$

Consequently, we update the current point, $(x, y)$, by taking the step $p_t$ to reach the next point. In our experiments, we employ Newton's method with a Wolfe linesearch, which is known to converge rapidly when initialized near a root, as shown in (Nocedal & Wright, 2006, Ch. 11). Note that the Jacobian $J_t$ requires minimal effort to compute in the lower-dimensional spaces that are classically amenable to black-box optimization. We provide exact implementation details in Appendix B.

**Adapting LLGAME:** We note that other optimizers can be used to solve for (first-order) LNPs of (5). One prevalent example is the gradient ascent-descent method (discussed by Mescheder et al. (2017) and Balduzzi et al. (2018), among others), which uses gradient steps instead of Newton steps as in our method. Our framework is flexible and LLGAME can readily be adapted to use these methods to identify local Nash points.

### 4.2    SOLVING THE HIGH-LEVEL GAME: FINDING LOCAL SADDLE POINTS WITH BSP

In the high-level game, we seek to solve zero-sum game (4) to identify the saddle points of $\mu_t$. Upon extracting a solution to (5) in LLGAME, we sample the objective $f$ around the low-level Nash point $(\bar{x}^*, \bar{y}^*)$. The result of this sampling is used to update the mean $\mu_t$ and uncertainty estimate $\sigma_t$ for the GP surrogate of $f$. Following the design of Sec. 4.1, we seek to identify first-order LSPs of (4) with roots of the function $G_t^f$ and global minima of the corresponding merit function $M_t^f$:

$$G^f(x, y) = \begin{bmatrix} \nabla_x f(x, y) \\ -\nabla_y f(x, y) \end{bmatrix}, \qquad M^f(x, y) = \frac{1}{2}\|G^f(x, y)\|_2^2. \quad (8)$$

However, as $f$ is unknown, we instead define function $G_t^{\mu}$ and corresponding merit function $M_t^{\mu}$ using the GP surrogate model of $f$ by replacing $f$ in (8) with $\mu_t$. Thus, our method identifies saddle points of $f$ by finding the roots of $G_t^{\mu}$ and global minima of $M_t^{\mu}$.

**The Bayesian Saddle Point Algorithm:** We now present our overall algorithm BSP (Alg. 2), which searches for the local saddle points. As stated earlier, this distinction between the two games allows us to confirm if a local Nash point is a local saddle point. In Alg. 2, we start by optimizing the hyperparameters of our GP kernel using the initial dataset, $\mathcal{S}_0$, a standard procedure in BO (Snoek et al., 2012). Upon learning the hyperparameters, an initial GP prior, $(\mu_0, \sigma_0)$, is obtained. Then in line 1, we select a starting point, $(x_0, y_0)$, from the initial dataset, $\mathcal{S}_0$, based on the lowest *merit* value. From this point, an iterative search for the local saddle point is conducted in the outer while loop (lines 2-6). The LLGAME function is utilized in line 3 to determine the subsequent point $(x_{t+1}, y_{t+1})$, which is a local Nash point of the general-sum game (5). The point, $(x_{t+1}, y_{t+1})$, is only a local Nash point for the given $\text{LCB}_t$ and $\text{UCB}_t$; we will not be sure if it is a local saddle point of $f$ until we sample $f$ at that point and calculate $M_t^{\mu}$ to validate the conditions in Prop. 3.3.

Consequently, in line 4, the point returned by LLGAME is sampled and added to the current dataset. In line 5, new hyperparameters are learned from dataset $\mathcal{S}_{t+1}$, and $\mu_{t+1}, \sigma_{t+1}, \mathrm{LCB}_{t+1}, \mathrm{UCB}_{t+1}$ are updated accordingly. [1] After sampling at the point $(x_{t+1}, y_{t+1})$, we have decreased the variance at that point, and thus the mean, $\mu_t(x_{t+1}, y_{t+1})$, and its gradient $\nabla \mu_t$ are better representations of $f$ and its gradient $\nabla f$. After each update to the GP surrogate, we check if the merit function value is sufficiently small, i.e. $M_t^\mu(x_t, y_t) \leq \epsilon$, where $\epsilon > 0$ is a user-specified tolerance. Ultimately in line 7, the local first-order saddle point $(x^*, y^*)$ is returned after the completion of the outer loop.

## 4.3 CONVERGENCE

In Lemma 4.1, we demonstrate that Alg. 1 will terminate and converge to a point which satisfies Prop. 3.3 under standard technical assumptions for Newton steps to be descent directions on the merit function. Our experiments in Sec. 5 include cases that both satisfy and violate these assumptions, showcasing the performance of our algorithm under various circumstances.

**Lemma 4.1** (Convergence to Local Nash Point in LLGAME). *(Nocedal & Wright, 2006, Thm. 11.6) Let $J(x, y)$ be Lipschitz continuous in a neighborhood $\mathbb{R}^{n_x} \times \mathbb{R}^{n_y} \subset \mathbb{R}^{k \times k}$ surrounding the sublevel set $\mathcal{L} = \left\{ x, y : M_t^{\mathrm{CB}}(x, y) \leq M_t^{\mathrm{CB}}(x_0, y_0) \right\}$. Assume that both $\|J(x, y)\|$ and $\|G_t^{\mathrm{CB}}(x, y)\|$ have upper bounds in $\mathbb{R}^{n_x} \times \mathbb{R}^{n_y}$. Let step lengths $\alpha_k$ satisfy the Wolfe conditions (Nocedal & Wright, 2006, Sec. 3.1). If $\|J(x, y)^{-1}\|$ has an upper bound, then Alg. 1 will converge and return a root of $G_t^{\mathrm{CB}}(x, y)$ which satisifes Prop. 3.3.*

In Alg. 2, we sample $f$ at the LNP returned by Alg. 1 to reduce the variance, $\sigma_t$, in the confidence bounds. This improves the accuracy of the mean function $\mu_t$ and its gradient $\nabla \mu_t$ as estimators for $f$ and its gradient $\nabla f$ around the sampled point. Consequently, the merit function, $M_t^\mu$, closely approximates $M^f$ and thus effectively validates convergence to the local (first-order) saddle point. As estimates improve with iterations, the loop in Alg. 2 is expected to terminate at a local saddle point.

Next, we provide an intuitive explanation of why sampling at local Nash points of confidence bounds in (5) will lead to local saddle points of $f$ in (1). As we sample local Nash points, the variance in the confidence bounds at those points will decrease, and LCB/UCB will get closer to each other, as shown by Lemma A.1 and Remark A.2. As we keep sampling, the variance will eventually become the noise variance, $\sigma_z$. As such, $\nabla \sigma \to 0$ and therefore $\nabla \mathrm{LCB}, \nabla \mathrm{UCB} \to \nabla \mu \to \nabla f$, and therefore finding local Nash points of the confidence bounds will eventually lead to first-order saddle points of $\mu$ and consequently of $f$. To confirm that we find a saddle, we verify the second-order condition (Prop. 3.4) for the final saddle point returned by Alg. 2. If this point does not satisfy the second-order conditions, we reinitialize our algorithm from a new initial point. We find that, in practice, our algorithms find LSPs on the first initialization more frequently than baseline methods.

## 4.4 VARIANTS OF BSP

**BSP Expensive:** Our BSP method in Alg. 2 aims to minimize queries of the function $f$ by taking multiple Newton steps per query of $f$. However, the algorithm may become unstable if the confidence bounds $\mathrm{UCB}_{t+1}, \mathrm{LCB}_{t+1}$ do not accurately approximate $f$. Additionally, it is possible that querying $f$ can often be inexpensive, for example, in the case of Reinforcement Learning in simulated environments (Sutton & Barto, 2018). In this case, we query $f$ during each iteration of Alg. 2 after taking a single Newton step, in contrast to the multiple steps taken in Alg. 1 (lines 2-3). This approach is referred to as BSP-*expensive* since we make more queries of $f$, while our original algorithm in Alg. 2 is referred to as BSP-*efficient*. We demonstrate in our results that this variant can more effectively and efficiently solve complex scenarios than baseline methods under these conditions.

**Exploration and Exploitation:** In Alg. 2, we encourage more exploration by using LCB for minimization and UCB for maximization. Since the value of unexplored regions has high variance and thus a lower LCB value, the minimization procedure will explore those regions first (vice-versa for UCB and maximization). As such, we refer to our original proposed method, as BSP-*explore*. Alternatively, we can use LCB for maximization and UCB for minimization, thus promoting more exploitation by our algorithm. We will refer to this variant as BSP-*exploit*. In real-world scenarios, this approach might be suitable when optimizing a well-understood process, fine-tuning known models, or when domain knowledge allows for confidently focusing on exploitation.

---

[1] This is a standard procedure in black-box optimization, explained in further detail in Appendix B.7.

## 5 EXPERIMENTS

In this section, we evaluate the BSP algorithms presented in Alg. 2 and Sec. 4.4 across various test environments. We consider four versions: 1) BSP-*efficient-explore* (EF-XPLORE), 2) BSP-*efficient-exploit* (EF-XPLOIT), 3) BSP-*expensive-explore* (EXP-XPLORE), and 4) BSP-*expensive-exploit* (EXP-XPLOIT). Our experiments, which prior work accepts as challenging baseline problems, demonstrate that each algorithm excels in specific settings. We consider the following three test cases.

**1. Decaying Polynomial:** In our first experiment, we examine the performance of our algorithms for a nonconvex-nonconcave objective taken from (Mazumdar et al., 2020; Gupta et al., 2024):

$$f_{\exp}(x, y) = \exp\left(-0.01\left(x^2 + y^2\right)\right)\left(\left(0.3x^2 + y\right)^2 + \left(0.5y^2 + x\right)^2\right). \tag{9}$$

This example is particularly difficult for three reasons: first, multiple LSPs exist. Second, the origin is a spurious saddle point which satisfies first-order conditions but not second-order conditions. Third, the function gradients decay to zero further from the origin, meaning that an iterative algorithm strays too far may "stall," taking smaller and smaller step sizes, but never converging to a fixed point.

**2. High-dimension Polynomial:** We consider a high-order polynomial from Bertsimas et al. (2010):

$$f_{\text{bertsimas}}(x, y) = -2x^6 + 12.2x^5 - 21.2x^4 - 6.2x + 6.4x^3 + 4.7x^2 - y^6 + 11y^5$$
$$-43.3y^4 + 10y + 74.8y^3 - 56.9y^2 + 4.1xy + 0.1y^2x^2 - 0.4y^2x - 0.4x^2y. \tag{10}$$

The decision space is within $[x_{\min} = -0.95, x_{\max} = 3.2] \times [y_{\min} = -0.45, y_{\max} = 4.4]$. The objective, $f_{\text{poly}}(x, y)$, is nonconvex-nonconcave and has multiple LSPs (Defn. 3.2). We form a high-dimension $(2n)$-D polynomial by letting, for $\vec{x} \in \mathbb{R}^n, \vec{y} \in \mathbb{R}^n$,

$$f_{\text{poly}}(\vec{x}, \vec{y}) = \sum_{i=1}^{n} f_{\text{bertsimas}}(x_i, y_i). \tag{11}$$

We set $n = 5$ to evaluate our proposed algorithms ability to identify LSPs in higher dimensions.

**3. ARIMA Tracking Model Predictive Controller (MPC):** Finally, we test our algorithms on a more realistic zero-sum game involving an ARIMA process that synthesizes a discrete time 1D time series of length $F$, denoted by $s_F \in \mathbb{R}^F$, for a model predictive controller to track. This setup mirrors real-world systems like that of Stent et al. (2024), in which an autonomous system corrects for distracted human driving. We represent the ARIMA process for initial state $s_0 \in \mathbb{R}$ and model parameters $\alpha \in \mathbb{R}, \beta \in \mathbb{R}$: $s_F = \text{ARIMA}(s_0, \alpha, \beta)$. The MPC takes the ARIMA time series, $s_F$, as input and returns a controller cost $f_{\text{MPC}} \in \mathbb{R}$ incurred while tracking the given time series. The MPC solves an optimization problem with quadratic costs and linear constraints, encapsulating vehicle dynamics and control limits. The optimization problem is represented as $f_{\text{MPC}} = \text{MPC}(A, B, s_0, s_F)$, returning the final overall cost of tracking the ARIMA-generated time series $s_F$, initial state, $s_0$ and model dynamics $A, B$. Further details can be found in Appendix B.5.

*Zero-Sum Game Formulation:* We formulate the interaction between the ARIMA forecaster and the MPC as a zero-sum game. The antagonist selects the ARIMA parameters $\alpha, \beta$ to generate difficult-to-track time series forecasts, $s_F$, resulting in a higher MPC cost $f_{\text{MPC}}$. In many scenarios, we want to find model dynamics that are robust and can effectively handle various tracking signals. As such, the protagonist chooses the MPC model dynamics parameters $A, B$ to accurately track $s_F$ and minimize the controller cost. This competitive scenario is formulated as follows:

$$f_{\text{MPC}} = \text{MPC}(\underbrace{A, B}_{\text{protagonist}}, \hat{s}_0, s_F = \text{ARIMA}(s_0, \underbrace{\alpha, \beta}_{\text{antagonist}})). \tag{12}$$

This game is motivated by real-world scenarios requiring robust controllers for adversarial and out-of-distribution inputs, and it has multiple LSPs (Defn. 3.2).

**Experimental Setup:** We compare our algorithms in two settings. In the first, we initialize our algorithms with a large number of sample points, modeling a scenario where the objective is well understood. In the second, we initialize algorithms with a small number of sample points, modeling a scenario where obtaining samples is expensive. In all our experiments, we use a squared exponential kernel, where the kernel value of two data points $x_i$ and $x_j$ is given by:

$$k(x_i, x_j) = \sigma_f^2 \exp\left[-\frac{1}{2}\frac{(x_i - x_j)^T(x_i - x_j)}{\sigma_l^2}\right], \tag{13}$$

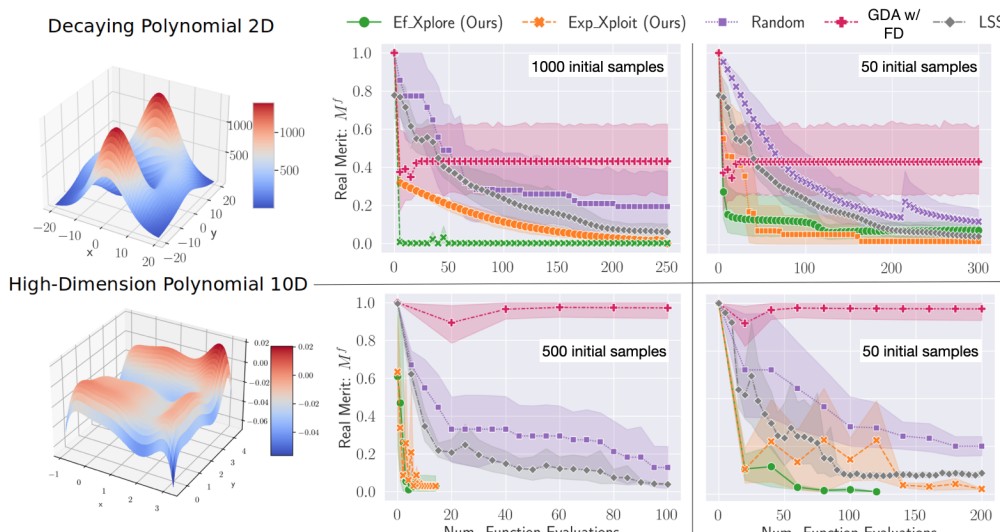

Figure 1: **Comparisons of selected algorithm variants with baselines:** We compare variants of our proposed algorithms with baseline methods across two domains (rows), the decaying and high-dimension polynomials, landscapes of which are shown in the first column. The middle column considers test cases with a large number of initially sampled points, while the right column examines test cases with a limited number of initially sampled points. In each case, we report the value of (real) merit function $M^f$ vs. the number of underlying function evaluations. *Key takeaway:* Generally, EF-XPLORE converges faster with a large number of initial samples by taking multiple Newton steps at each step in order to exploit the accurate prior while EXP-XPLOIT exhibits quicker convergence with limited samples by taking single Newton steps to avoid unfavorable regions amid uncertainty. Finally, we find that EF-XPLORE and EXP-XPLOIT converge faster than all three baseline methods, indicating the benefit of the GP surrogate in improving convergence compared to baselines which are often unable to converge.

where $\sigma_f$ (signal variance) and $\sigma_l$ (signal length scale) are hyperparameters. We learn these hyperparameters via maximum likelihood to initialize our algorithms. In all experiments, we assume that we observe noisy measurements of the underlying function. To ensure that algorithms are initialized at points with a non-zero gradient in the decaying polynomial example (top row), we ensure these are selected from the non-flat regions of the function. All experiments are performed with 20 seeds for each algorithm. For the scenario with a limited number of initially sampled points, we sampled 50 points for the decaying and high-dimension polynomial objectives, and 10 points for the ARIMA-MPC objective. We present the exact experimental setup for each test case in Appendix B.

## 5.1 EXPERIMENTAL RESULTS

Fig. 1 summarizes the performance of the EF-XPLORE and EXP-XPLOIT variants of our algorithms for the first two scenarios mentioned above; for results related to EF-XPLOIT and EXP-XPLORE, we refer the reader to Fig. 4 in Appendix C, where we report similar results to EF-XPLORE and EXP-XPLOIT, respectively. The middle column of Fig. 1 considers test cases with a large number of initially sampled points, while the right column considers test cases with a limited number of initially sampled points. We sample 1000 initial points for the decaying polynomial and 500 initial points for the high-dimensional polynomial and for ARIMA-MPC (for which we report results in Fig. 2).

We assess the performance of our algorithms using the merit function from (8), $M^f$, which is calculated based on the true gradients of the underlying function, rather than the confidence bounds employed in the actual algorithm. As previously mentioned, $M^f$ will attain a global minimum (of 0) when the first-order conditions in Prop. 3.3 are satisfied. This evaluation metric offers a direct measure of the algorithms' effectiveness in identifying local saddle points of the underlying function.

**Baselines:** We consider three baseline algorithms: naive random sampling (Random), gradient descent-ascent with finite differencing (GDA with FD), and local symplectic surgery (LSS), a state-of-the-art baseline from Mazumdar et al. (2019). These methods assume access to zeroth-, first-, and second-order derivatives, respectively. In the random sampling baseline, we uniformly sample a fixed

number of points $(x, y)$ from the hyperbox $\{x, y : x_{\min} \leq x \leq x_{\max}, y_{\min} \leq y \leq y_{\max}\}$, and retain the point with the lowest real merit function value ($M^f$). In the gradient descent-ascent baseline, we employ finite differencing to estimate each player's gradient and use a Wolfe linesearch to select step sizes. This approach allows for a more directed search compared to random sampling. Approximating the gradient with finite differencing provides a fair comparison between our method and a first-order approach using zeroth-order samples. For the decaying and high-dimension polynomial examples, we compare with LSS. LSS requires access to function gradients and Hessians; rather than provide finite differenced estimates (which can be extremely noisy and require excessive function evaluations), we provide oracle access to the true function derivatives and corrupt with standard Gaussian noise.

**Analysis of a Large Number of Initially Sampled Points:** In this setting, we observed that the *exploit* variants of our proposed algorithms, EF-XPLOIT (blue) and EXP-XPLOIT (orange), demonstrated the fastest convergence. This outcome is expected since the accuracy of the confidence bounds was higher, reducing the need for exploration. Overall, EF-XPLOIT (blue) achieved the fastest convergence in these experiments due to its ability to take multiple accurate Newton steps. In the decaying polynomial example from Fig. 1, EF-XPLORE converges quickly as well as taking single Newton steps helps the algorithm converge in this particularly complicated landscape. In contrast, the *explore* algorithms had slower convergence, as they prioritize exploration. This general pattern continues to hold in Fig. 2, for the ARIMA-MPC scenario.

**Analysis of Limited Number of Initially Sampled Points:** In this setting, we observed that the *explore* variants of our proposed algorithms, EF-XPLORE (green) and EXP-XPLORE (red), achieved the best performance. Notably, the algorithm variant EXP-XPLORE (red) demonstrated fast convergence, which can be attributed to its exploration approach and avoiding multiple incorrect Newton steps in the face of uncertainty. The *exploit* variants, EF-XPLOIT (blue) and EXP-XPLOIT (orange) exhibited slower convergence, as they relied too heavily on prior information and consequently took incorrect steps. Specifically, the EF-XPLOIT (blue) variant failed to converge for some seeds since it took incorrect Newton steps and was unable to explore. Finally, the *expensive* variants, in general, are more stable in this setting, as they only take single Newton steps and are less likely to reach unfavorable regions. EXP-XPLOIT (orange), for example, converges to a real merit vaue $M^f = 0$ for both experiments. We note that for the decaying polynomial example, we still see the efficient variants converge faster, and this result reflects the complexity of the objective landscape.

**Comparisons with Baselines:** Random sampling and GDA with FD never converge, though random sampling reduces the merit function value. The noise introduced by finite differencing renders GDA with FD ineffective in this problem setting. Despite reaching a low merit function value, we note that LSS fails to converge in many of these scenarios. In the decaying polynomial example, we see LSS iterate outward far from the origin where $\nabla f$ becomes very small and no saddle points exist. Note that this behavior is consistent with Mazumdar et al. (2019), which claims only that if LSS converges, it finds a saddle. In Appendix C, our experiments on the decaying and high-dimension polynomials indicate that LSS fails to converge to a saddle point far more often than BSP (Table 2).

**Runtime and Success Rate:** Neglecting the time to query the underlying function, the *efficient* variants require 400-500% more time compared to the *expensive* variants, as they take multiple Newton steps between each iteration or underlying function sample. Additionally, the EF-XPLOIT (blue) variant did not achieve a 100% success rate in cases with a limited number of initial samples, as it relied on exploitation and took incorrect Newton steps due to high uncertainty. We note that LSS often fails to converge to a LSP due to algorithmic assumptions (which only guarantee that when LSS converges, it finds a saddle) or domain-specific factors (iterating towards low-gradient regions of the objective landscape). We find that EXP-XPLOIT, EXP-XPLORE, and EF-XPLORE find saddle points strictly more often than LSS, and that EF-XPLOIT performs similarly or better depending on the experiment. We provide a detailed comparison of convergence and success rates in Appendix C.

**Key takeaways from our experimental results:** Our experimental results highlight several important insights about our algorithms variants. 1) The *efficient* variants require fewer underlying function evaluations and will work best when the prior is accurate. 2) The *expensive* variants offer faster runtimes and will provide more stable convergence when the prior is inaccurate. 3) The *explore* variants provide a higher success rate when the number of initial samples is limited. 4) The *exploit* variants exhibit faster convergence in the setting with a large number of initial samples. These findings suggest that the choice of an algorithm variant should depend on the specific characteristics of the problem at hand, such as the number of available initial samples, runtime requirements, and

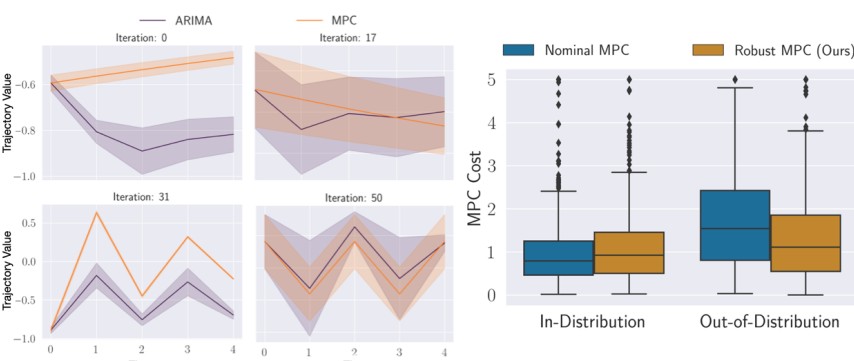

Figure 2: **Saddle Point Optimization with BSP leads to Robust MPC on Out-of-Distribution (OoD) data:** On the left, we display the MPC tracking of the timeseries generated by the ARIMA model at various iterations for the EF-XPLORE variant. The ARIMA target trajectory is depicted in purple, while the corresponding MPC tracking is illustrated in orange. Initially, MPC performs poorly (iteration 0), but gradually improves its tracking (iteration 17). Consequently, the ARIMA makes tracking more challenging for the MPC (iteration 31), until they both reach equilibrium (iteration 50). On the right, we compare the final robust MPC parameters (orange), obtained through our algorithm, to the nominal MPC parameters (blue) on in-distribution data (left column) and OoD data (right column). *Key takeaway*: significantly, the robust MPC successfully identifies robust MPC parameters and achieves **27.6%** lower mean MPC cost on OoD data compared to nominal MPC without reducing performance on in-distribution data.

domain knowledge of the underlying objective. Moreover, we find that our methods converge faster than simple baselines based on random sampling and finite differencing, and that they converge faster (and more reliably) than the state-of-the-art LSS algorithm when it is adapted to the black-box setting.

**Performance of BSP Variants in ARIMA-MPC Example:** In Fig. 2, we evaluate the performance of the MPC parameters found by our algorithms. Specifically, we compare the performance of the MPC parameters obtained at the end of the EF-XPLORE variant on both the in-distribution and Out-of-Distribution (OoD) ARIMA forecasting timeseries datasets. When MPC operates at a LSP, we can expect it to be robust to perturbations in ARIMA parameters and therefore to OoD time series. Indeed, controller parameters found by our algorithm achieve 27.6% lower mean MPC cost, thus showcasing our algorithm's ability to locate saddle points which correspond to robust performance on OoD data without reducing performance on in-distribution data. We provide the exact details of this experiment in Appendix B, and present full convergence results on the ARIMA-MPC example in Fig. 4. These results mirror those of previous experiments, and we find that the exploit variants converge faster with many initial samples while the efficient variants do so with limited initial samples.

**Summary:** Our experimental results conclusively indicate that our proposed BSP algorithms converge faster, sample more efficiently, and produce more robust solutions than existing methods in a variety of black-box saddle point optimization problems.

## 6 CONCLUSION

We present a BO-inspired framework for identifying local saddle points for an unknown objective function, $f$, with zeroth-order samples. We frame the problem of finding local saddle points for an unknown objective function as a two-level procedure. A *low-level* algorithm constructs a general-sum game from a Gaussian process which approximates the unknown function $f$, and solves for the local Nash points of this game by finding the roots of a system of nonlinear algebraic equations. A *high-level* algorithm queries the points returned by the *low-level* algorithm to refine the GP estimate and monitor convergence toward local saddle points of the original problem. We validate the effectiveness of our algorithm through extensive Monte Carlo testing on multiple examples.

**Limitations and Future Work:** While our proposed framework shows promising results, we plan to address certain limitations in future work. First, we intend to directly incorporate second-order sufficient conditions (Prop. 3.4) to enhance the performance of our approach within the general-sum low-level game. Second, to further demonstrate our framework's adaptability, we will test our algorithm with other acquisition functions, such as knowledge gradient (Ryzhov et al., 2012) and entropy search (Hennig & Schuler, 2012).

**Reproducibility Statement.** The pseudo-code and hyper-parameter details have been provided to help reproduce the results reported in the paper. The source code will be released post publication.

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

## A   TECHNICAL INSIGHTS

In this section, we delve into the convergence properties of our proposed algorithms. As mentioned in Sec. 4.3, when we sample local Nash points obtained by Alg. 1, the variance in the confidence bounds at that local Nash point decreases, and LCB/UCB converge towards each other eventually at the local Nash point. In Lemma A.1, we demonstrate that with zero observation noise, the confidence bounds at a sampled point are equal, i.e., LCB = UCB. Subsequently, in Remark A.2, we explore the more general case involving non-zero observation noise, positing that as we repeatedly sample in close proximity to the same point, the variance of the sampled point becomes predominantly dependent on the observation noise, resulting in LCB ≈ UCB. Lastly, in Appendix A.2, we provide experimental evidence to corroborate the decrease in the variance of the sampled point during our algorithm's execution.

Consequently, these results attest that as we sample the local Nash point, the variance in the GP at that point will decrease, and eventually becomes a observation noise variance. As such, the gradient of the variance, $\nabla \sigma \to 0$, and therefore $\nabla \text{LCB}, \nabla \text{UCB} \to \nabla \mu$. As stated in Sec. 4.3, since $\nabla \mu$ will become a reliable estimator of the true gradients of the unknown function, $\nabla f$, finding local Nash points of the confidence bounds will eventually lead to local saddle points of $\mu$ and consequently of $f$. This convergence property is a key feature of our proposed algorithms, ensuring that the method converges to a solution that represents a local saddle point of the underlying unknown function.

### A.1   UCB AND LCB WILL APPROACH ONE ANOTHER AT SAMPLED POINTS

For the ease of the proofs, we will focus on the case when $r = f(x) + z$, where $z \sim \mathcal{N}\left(0, \sigma_z^2\right)$. The proof can easily be generalized for $r = f(x, y) + z$.

We denote the set of observed points as $\mathbf{X} = \{x_1, x_2, \ldots, x_n\}$ and $r = \{r_1, r_2, \ldots, r_n\}$. Consider that the point, $x_*$, has already been sampled, as such $x_* = x_i$ for some $i \in \{1 \ldots N\}$. Now, recall the predictive variance of the point $x_*$ is:

$$\sigma(x_*|\mathbf{X}, r, x_*) = K\left(x_*, x_*\right) - k_*^\top \left(\Sigma(\mathbf{X}, \mathbf{X}) + \sigma_z^2 I\right)^{-1} k_*, \tag{14}$$

where $k_* = [K(x_1, x_*), K(x_2, x_*), \ldots, K(x_n, x_*)]^\top$, and $\Sigma(\mathbf{X}, \mathbf{X}) \in \mathbb{R}^{n \times n}$ given by

$$
\begin{bmatrix}
K(x_1, x_1) & K(x_1, x_2) & \cdots & K(x_1, x_n) \\
\vdots & \vdots & \ddots & \vdots \\
K(x_n, x_1) & K(x_n, x_2) & \cdots & K(x_n, x_n)
\end{bmatrix}. \tag{15}
$$

**Lemma A.1** (Equality of $\text{UCB}_t$ and $\text{LCB}_t$ at sampled points under zero observation noise). *Consider the upper confidence bound, $\text{UCB}_t$, and the lower confidence bound, $\text{LCB}_t$, for a given time step $t$. In the case of zero observation noise, i.e., $z_t = 0$, the confidence bounds become equal for any sampled point $x$ such that $\text{UCB}_t(x) = \text{LCB}_t(x)$.*

*Proof.* Since, $z_t = 0$, the predictive variance at the point $x_*$ is:

$$
\sigma(x_* | \mathbf{X}, r, x_*) = K(x_*, x_*) - k_*^\top \Sigma^{-1} k_*. \tag{16}
$$

Let the point $x_*$ be some point $x_i \in \mathbf{X}$, i.e., $x_* = x_i$ for some $1 \leq i \leq n$. Then, the corresponding kernel vector is given by the $i$-th column of the covariance matrix $\Sigma(\mathbf{X}, \mathbf{X})$, so $k_* = k_*^\top = \Sigma_{:,i}$ and the kernel value is $K(x_*, x_*) = K(x_i, x_i)$. The variance at the new point $x_*$ becomes:

$$
\sigma(x_* | \mathbf{X}, r, x_*) = K(x_i, x_i) - \Sigma_{:,i}^\top \Sigma^{-1} \Sigma_{:,i}. \tag{17}
$$

Next, we have:

$$
\Sigma^{-1} \Sigma_{:,i} = \mathbf{e}_i, \tag{18}
$$

where $\mathbf{e}_i$ is the $i$-th standard basis vector. This can be seen from the property of the inverse matrix, i.e., $\Sigma(\mathbf{X}, \mathbf{X})^{-1} \Sigma(\mathbf{X}, \mathbf{X}) = I$, where $I$ is the identity matrix.

Thus, the variance at the new point $x_*$ simplifies to:

$$
\sigma(x_* | \mathbf{X}, r, x_*) = K(x_i, x_i) - \Sigma_{:,i}^\top \mathbf{e}_i = K(x_i, x_i) - \Sigma_{i,i}. \tag{19}
$$

Since $x_*$ is a previously sampled point, the kernel function $K(x_i, x_i)$ and $\Sigma_{i,i}$ are equal to 1. Thus, the variance at $x_*$ is:

$$
\sigma(x_* | \mathbf{X}, r, x_*) = 1 - 1 = 0. \tag{20}
$$

This shows that the variance at a previously sampled point $x_*$ is zero in the no observation noise case, for any kernel function.

UCB and LCB at the point $x_*$ are given by:

$$
\text{UCB}_t(x_*) = \mu_t(x_*) + \beta_t \sigma_t(x_*), \quad \text{LCB}_t(x_*) = \mu_t(x_*) - \beta_t \sigma_t(x_*). \tag{21}
$$

Since we just showed the sampled point, $x_*$, the variance $\sigma_t(x_*) = 0$. Then:

$$
\text{UCB}_t(x_*) = \text{LCB}_t(x_*) = \mu_t(x_*) \tag{22}
$$

$\square$

*Remark* A.2 (Approximate equality of UCB and LCB at sampled points under observation noise). Consider the upper confidence bound, $\text{UCB}_t$, and the lower confidence bound, $\text{LCB}_t$, for a given time step $t$. In the presence of observation noise, i.e., $z_t \sim \mathcal{N}(0, \sigma_z^2)$, when the same point is sampled repeatedly, or nearby points are sampled, the predictive variance $\sigma(x)$ approaches the observation noise $\sigma_z^2$. This is based on the fact after repeated sampling, the only uncertainty left about the sampling point will be due to the observation noise. As such, as the predictive variance becomes smaller due to repeated sampling, the confidence bounds at the sampled point $x$ will have $\text{UCB}_t(x) \approx \text{LCB}_t(x)$.

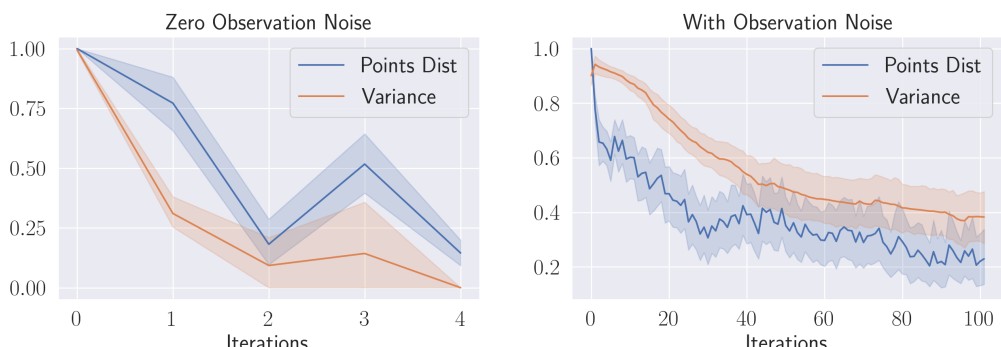

Figure 3: **The variance at the sampled points decreases over time.**

### A.2    Variance of the Sampled Point

In Fig. 3, we demonstrate the reduction in the variance of the sampled points as we approach the saddle point during the execution of our algorithms. We compare cases with and without observation noise for a convex-concave objective function of the form $ax^2 + bxy - cy^2$, where coefficients $a, b, c > 0$, scenario using 10 seeds. We note that LSPs will be locally convex-concave in their neighborhoods and so these results will apply locally in those scenarios. The blue line represents the distance between two consecutive points for each function evaluation, while the orange line indicates the variance in the GP prior at the sampled points. The primary observation is that as we take smaller Newton steps and sample points close to each other, the variance at those points decreases, thus increasing the accuracy of the mean function $\mu_t$ and its gradient $\nabla \mu_t$ as estimators for the objective function $f$ and its gradient $\nabla f$ around the sampled points.

## B    Experimental Details

We provide the exact implementation details of all the experiments. Starting with compute, all experiments were conducted on a desktop computer equipped with an AMD Ryzen 9 5900X CPU and 32 GB RAM. No GPUs were required for these experiments.

### B.1    Newton's method implementation details

**1. Invertibility of Jacobian:** To solve for the Newton step $p(x_t, y_t)$ in (7), the Jacobian matrix $J(x_t, y_t)$ must be non-singular. Therefore, at each new iterate, we need to verify the invertibility of $J(x_t, y_t)$. A common way to ensure Hessian invertibility is by adding a constant factor $\lambda I$ to the diagonal. Gill & King (2004) offers a concise overview of alternative methods for inverting the Hessian matrix.

**2. Line-search:** Newton's method alone (with a unit step length) does not guarantee convergence to the root unless the starting point is sufficiently close to the solution. To enhance robustness, we employ line-search, using the merit function $M$ as the criterion for sufficient decrease. The use of line search is standard practice, as discussed and explained in (Nocedal & Wright, 2006, Ch.3).

### B.2    Local Symplectic Surgery implementation details

For LSS, we utilize the same regularization and parameters described by (Mazumdar et al., 2019, Sec. 5.1). We use ForwardDiff.jl (Revels et al., 2016) for computing gradients for LSS.

### B.3    Decaying polynomial implementation details

The objective function of the decaying polynomial problem is depicted in the top left plot of Fig. 1. The decision variables were $x, y \in \mathbb{R}^2$. To ensure that algorithms are initialized at points with a non-zero gradient in the decaying polynomial example (top row), we ensure these are selected from

the non-flat regions of the function (a distance between 9 and 18 from the origin). The Hessian regularization constant was $\lambda = 0.01$. The strong Wolfe parameters, according to (Nocedal & Wright, 2006, Ch.3), were $c_1 = 0.01$ and $c_2 = 0.7$. We added observation noise $z_t \sim \mathcal{N}\left(0, \sigma_z^2 = 1\right)$ to each underlying objective function sample. We utilize JAX (Bradbury et al., 2018) to compute gradients for this example.

## B.4 HIGH-DIMENSION POLYNOMIAL IMPLEMENTATION DETAILS

The objective function of the high-dimension polynomial problem is depicted in the bottom left plot of Fig. 1. The decision variables were $x, y \in \mathbb{R}^5$, resulting in a combined decision variable of 10 dimensions. The Hessian regularization constant was $\lambda = 0.01$. The strong Wolfe parameters, according to (Nocedal & Wright, 2006, Ch.3), were $c_1 = 0.01$ and $c_2 = 0.7$. We added observation noise $z_t \sim \mathcal{N}\left(0, \sigma_z^2 = 0.003\right)$ to each underlying objective function sample. We utilize JAX (Bradbury et al., 2018) to compute gradients for this example.

## B.5 ARIMA TRACKING MODEL PREDICTIVE CONTROLLER (MPC)

In this experiment, an ARIMA process synthesizes a discrete time $1D$ time series of length $F$, denoted by $s_F \in \mathbb{R}^F$, for an MPC controller to track. Specifically, the ARIMA process generates the time series as: $s_{t+1} = \mu + \alpha s_t + \beta w_{t-1} + w_t$, where $\mu \in \mathbb{R}$ is the mean, $w = \mathcal{N}(0, \sigma) \in \mathbb{R}$ is noise, and $\alpha \in \mathbb{R}$, $\beta \in \mathbb{R}$ are model parameters. Consequently, we represent the ARIMA process for initial state $s_0$ and model parameters $\alpha, \beta$: $s_F = \mathrm{ARIMA}(s_0, \alpha, \beta)$.

The MPC controller takes the ARIMA time series, $s_F$, as input and returns a controller cost $f_{\mathrm{MPC}} \in \mathbb{R}$ to track the given time series. The MPC controller solves the following optimization problem:

$$\min_{\hat{u}} \mathrm{MPC}(A, B, \hat{s}_0, s_F) = \sum_{t=0}^{F} \left(\hat{s}_t - s_t\right)^\top Q \left(\hat{s}_t - s_t\right) + \hat{u}_t^\top R \hat{u}_t. \tag{23a}$$

$$\text{subject to: } \hat{s}_t = A\hat{s}_{t-1} + B\hat{u}_{t-1} \tag{23b}$$

$$u_{\min} \leq u_t \leq u_{\max}. \tag{23c}$$

The optimization problem has quadratic costs and linear constraints. The quadratic costs in (23a), measure how well the controller tracks the timeseries $s_F$, and how much controller effort was used. The linear constraints in (23b) encapsulate the system dynamics, and $A, B \in \mathbb{R}$ are controller parameters that describe the dynamics. The MPC has additional control constraints in (23c), which describe the control limits of the controller. As such, we represent the optimization problem of the MPC controller as follows: $f_{\mathrm{MPC}} = \mathrm{MPC}(A, B, \hat{s}_0, s_F)$, which returns the final overall cost of tracking the ARIMA-generated time series $s_F$, and initial state, $\hat{s}_0$.

In our experiment, the decision variables were $x, y \in \mathbb{R}^2$, which resulted in a combined decision variable of four dimensions. We set the Hessian regularization constant to $\lambda = 0.001$. Following the recommendations in (Nocedal & Wright, 2006, Ch.3), we chose strong Wolfe parameters $c_1 = 0.01$ and $c_2 = 0.8$. Both the parameters of the ARIMA process $(\alpha, \beta)$ and the MPC parameters $(A, B)$ were constrained to lie within the range $[-1, 1]$. By incorporating these constraints and parameters, we ensured a consistent framework for the optimization problem while providing sufficient flexibility for the ARIMA process and the MPC to interact in the zero-sum game. We utilized CVXPYLAYERS (Agrawal et al., 2019) for computing gradients.

## B.6 ROBUST MPC EXPERIMENTAL DETAILS

In this section, we provide the details of the Robust MPC experiments. In this experiment, we demonstrated that the MPC parameters found at the end of our zero-sum game between the ARIMA antagonist player and the MPC protagonist player will be more robust to out-of-distribution data. Our algorithm will converge to a local saddle point of this zero-sum game, where both players will be in equilibrium. The ARIMA antagonist player cannot find more adversarial parameters for MPC, while MPC cannot get better at tracking ARIMA generated forecasts.

Specifically, for the experiment, we compared a nominal MPC with a robust MPC (found using our method) on in-distribution data and out-of-distribution data. To generate in-distribution data, we

sampled 500 ARIMA time series forecasts, $s_F$, with $\alpha, \beta \in [-1, 1]$. We chose out-of-distribution ARIMA parameters similar to the final ARIMA antagonist player parameters. Although the parameters were constrained to be within $[-1, 1]$ during the actual algorithm, we selected out-of-distribution parameters $\alpha = -0.1$ and $\beta = -1.2$ and sampled 500 ARIMA time series forecasts for these parameters. This choice enabled us to evaluate the robustness of the MPC against data that deviates from its original training distribution. Finally, we picked nominal MPC parameters, $A, B$, by fitting the MPC parameters to in-distribution data using supervised learning, i.e., the best $A, B$ to minimize the MPC tracking cost for the in-distribution data. We chose robust MPC parameters as the final MPC protagonist player at the convergence of the zero-sum game. By comparing the performance of the nominal and robust MPCs on both in-distribution and out-of-distribution data, we aimed to demonstrate the effectiveness of our method in finding robust MPC parameters that can handle deviations from the original training data distribution better than the nominal MPC.

In Fig. 2, we compared the MPC tracking costs of both the MPCs on in-distribution data and out-of-distribution data. As expected, the nominal MPC performs better on in-distribution data since it is trained on this data. However, the robust MPC significantly outperforms the nominal MPC on out-of-distribution data by achieving a $27.6\%$ lower mean MPC cost. Additionally, the poor performance of the nominal MPC indicates that the final ARIMA antagonist player parameters are indeed challenging. These results suggest that our algorithm has successfully identified robust MPC parameters and adversarial ARIMA parameters.

### B.7    UPDATING HYPERPARAMETERS

Steps for updating $\mu_{t+1}$, $\sigma_{t+1}$, $\text{UCB}_{t+1}$, and $\text{LCB}_{t+1}$:

1. Learn hyperparameters of the kernel function using maximum likelihood estimation, as explained in Section 2.3 of Rasmussen (2003).
2. Using the learned hyperparameter and updated kernel function, construct the $\mu_t$, $\sigma_t$ using the current dataset, as explained in Eq 2.25 and 2.26 in Rasmussen (2003).
3. Construct new $\text{UCB}_t$ and $\text{LCB}_t$. This is straightforward since:

$$\text{UCB}_t = \mu_t + \beta * \sigma_t, \text{LCB}_t = \mu_t - \beta * \sigma_t.$$

4. Collect new samples and repeat the process.

## C    ABLATION STUDIES

We generate full runtime, success rate, and convergence rate results for all of our algorithm variants across the decaying polynomial, high-dimension polynomial, and ARIMA-MPC examples. For a full discussion on our variants, we refer the reader to Sec. 5.

### C.1    RUNTIME COMPARISON

We can compare the runtimes of each algorithm variant in terms of the total number of Newton steps taken. In Table 1, we present the total number of Newton steps taken by each algorithm variant over 10 seeds for scenarios with a limited number of initially sampled points. The results demonstrate that the *expensive* variants require significantly fewer steps to converge compared to the *efficient* variants. This outcome is expected, as *expensive* variants only take one Newton step between each sample of the underlying function evaluation. Furthermore, as anticipated, the *explore* variants require more Newton steps due to their exploration approach, but the difference is not substantial.

### C.2    SUCCESS RATE COMPARISON

In Table 2, we display the percentage of successful seeds out of 20 seeds for each algorithm variant in scenarios with a limited number of initially sampled points. The results reveal that the *explore* variants exhibit more reliable convergence compared to the *exploit* variants. This outcome is expected, as *explore* variants emphasize exploration and, therefore, are more likely to converge. Additionally, the *expensive* variants demonstrate greater stability in this setting, as they only take single Newton steps and are less prone to reaching unfavorable regions. The EF-XPLOIT variant exhibits the lowest success rate in convergence, as it relies on exploitation and may take incorrect Newton steps. *We*

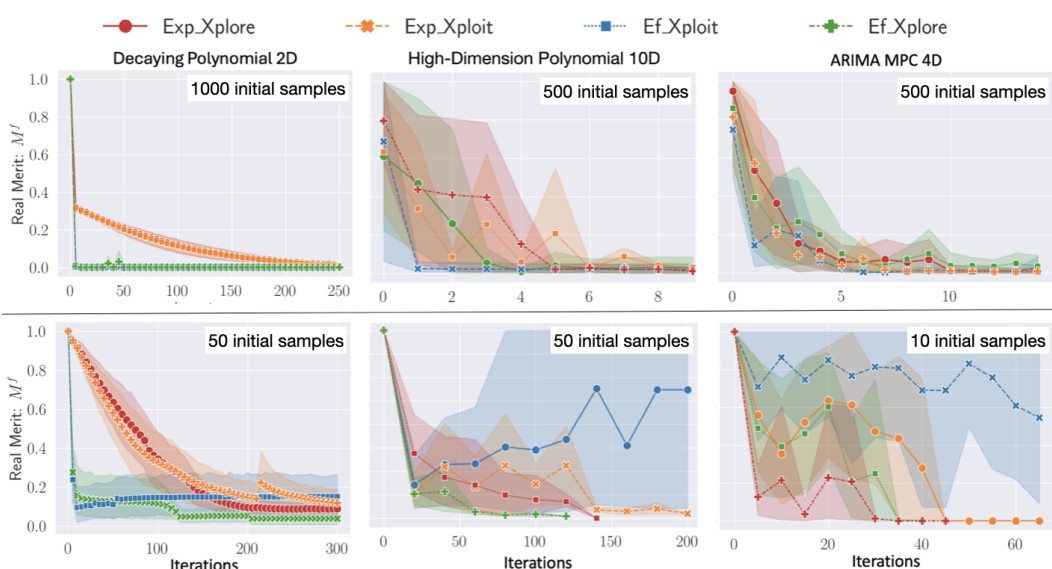

Figure 4: **Comparisons of our proposed algorithm variants:** We compare all four variants of our proposed algorithms across the three experiments (each column) described in Sec. 5. The horizontal axis denotes the number of underlying function evaluations, while the vertical axis represents the value of the real merit function, $M^f$. The top row considers test cases with a large number of initially sampled points, while the bottom row examines test cases with a limited number of initially sampled points. The *key takeaway* is that generally, *exploit* variants converge faster with a large number of initial samples due to effective utilization of accurate priors, while *explore* variants exhibit quicker convergence with limited samples by prioritizing exploration amid uncertainty. *Efficient* variants converge faster with many initial samples by taking multiple accurate Newton steps, while *expensive* variants show stable though often slower convergence with limited samples, taking single Newton steps to avoid unfavorable regions.

| Domain | Type of steps | Efficient | Expensive |
|---|---|---|---|
| High-Dimension Polynomial | Explore | 2858 | 1464 |
| | Exploit | 2613 | 1381 |
| Decaying Polynomial | Explore | 2690 | 1893 |
| | Exploit | 3000 | 3000 |
| ARIMA-MPC | Explore | 1053 | 353 |
| | Exploit | 815 | 229 |

Table 1: **Runtime Comparison:** In this table, we show the number of Newton steps taken by each algorithm variant for the limited number of initially sampled points scenarios over 10 seeds.

*include a run as a success if the BSP solution satisfies second-order sufficient conditions according to the ground truth derivatives* $\nabla f$ *and* $\nabla^2 f$. Lastly, we include success rates for when LSS, adapted to the black box setting, converges.

| Domain | Type of steps | Efficient | Expensive | Baseline |
|---|---|---|---|---|
| Decaying Polynomial | Explore | 60% | 60% | – |
| | Exploit | 30% | 50% | – |
| | LSS | – | – | 15% |
| High-Dimension Polynomial | Explore | 95% | 100% | – |
| | Exploit | 65% | 80% | – |
| | LSS | – | – | 70% |
| ARIMA-MPC | Explore | 90% | 95% | – |
| | Exploit | 75% | 85% | – |

Table 2: **Success Rate:** In this table, we show the percent of successful seeds out of 20 seeds for each algorithm variant for the limited number of initially sampled points scenarios. We show results on the LSS algorithm for two of the experiments, where we adapt LSS to the black box setting by providing an oracle to sample derivative information. To avoid including spurious saddle points as successes, we report the success rate according to the true second order conditions, defined in Prop. 3.4, at the first solution BSP finds (i.e., without reinitialization).

