# OpenReview forum: "A Framework for Finding Local Saddle Points in Two-Player Zero-Sum Black-Box Games"
_ICLR.cc/2025/Conference — Submitted to ICLR 2025_

### Official Review · Reviewer_euUW · 2024-10-18

**Soundness:** 3
**Presentation:** 3
**Contribution:** 3
**Rating:** 6
**Confidence:** 3

**Summary:**

This paper presents a novel framework for finding local saddle points in two-player zero-sum black-box games. The authors propose a bilevel approach that combines Gaussian processes to model the unknown, nonconvex-nonconcave objective and a lower-level game to identify sampling locations.

**Strengths:**

1. The framework innovatively combines ideas from Bayesian optimization with saddle point optimization in a black-box setting. It offers a zeroth-order technique for finding saddle points.
2. The explanation of the bilevel optimization framework is well-organized.
3. The ability to identify saddle points in nonconvex-nonconcave black-box settings is highly relevant for a wide range of applications, particularly those involving adversarial dynamics and game theory.
4. The experimental results are comprehensive, and the ablation studies have been conducted.

**Weaknesses:**

1. The theoretical guarantees of the algorithm in this paper are somewhat insufficient, with only Lemma 4.1 used to show that the algorithm converges to a local Nash point, and the assumptions made are rather strong.
2. The paper lacks a dedicated "Related Work" section. It appears that the related work has been split into two parts, placed separately in Section 1 and Section 2.

**Questions:**

1. Could you provide more theoretical analysis of the algorithm? The current analysis is somewhat limited, and there is no discussion on the rationale behind the assumptions in Lemma 4.1. Please offer an explanation here.
2. I suggest reorganizing Section 1 and Section 2. The current structure is somewhat disorganized and lacks a systematic introduction to prior work.
3. How to incorporate second-order sufficient conditions to enhance the performance of your approach? Please gives further explanation.

---

> ### Author Response · Authors · 2024-11-18
> **Response by Authors [1/6]**
>
> We thank the reviewer for their detailed comments. Please find our responses below. All references refer to works cited in the manuscript.
>
> **Comment:** The framework innovatively combines ideas from Bayesian optimization with saddle point optimization in a black-box setting. It offers a zeroth-order technique for finding saddle points. The explanation of the bilevel optimization framework is well-organized. The ability to identify saddle points in nonconvex-nonconcave black-box settings is highly relevant for a wide range of applications, particularly those involving adversarial dynamics and game theory. The experimental results are comprehensive, and the ablation studies have been conducted.
>
> **Response:** We thank the reviewer for their kind words for the strengths of our contribution, and for their time and effort in reviewing our manuscript.

---

> ### Author Response · Authors · 2024-11-18
> **Response by Authors [2/6]**
>
> **Comment:** The theoretical guarantees of the algorithm in this paper are somewhat insufficient, with only Lemma 4.1 used to show that the algorithm converges to a local Nash point, and the assumptions made are rather strong.
>
> **Response:** We thank the reviewer for their feedback on this important topic. As noted in our top-level comment, this manuscript opens up a new line of work in solving black-box zero-sum games. While we acknowledge that our algorithms may converge to a point which is not precisely a LNP, our empirical success rate results shown in Appendix Table 2 demonstrate that our algorithms find LSPs more frequently than a state-of-the-art baseline method from the literature (called LSS) even when it is provided with a gradient oracle. In the case of LSS, this is because the theoretical guarantee that LSS’ limit points coincide with local saddle points only holds if LSS converges. In practice, we find that LSS often fails to converge on common experiments used in the literature, as reported in Appendix Table 2. Thus, we believe that these empirical results are enough to establish the practical potential of the proposed black-box saddle point approach.
>
> In addition to Lemma 4.1, we highlight additional theoretical results Lemma A.1 and Remark A.2 which discuss how the LCB and UCB approach one another as more function evaluations are taken in subsequent iterations, meaning that the general-sum LLGame becomes more similar to the underlying zero-sum game as described in Section 4.
>
> **Manuscript Changes:**
> - Section 2: We provide a clearer discussion of the drawbacks of existing methods.
> - Section 4.3: We more directly reference Lemma A.1 and Remark A.2 from the Appendix which discuss the convergence of LCB and UCB under certain conditions.
> - Section 5: We clarify that our chosen experiments are drawn from example problems in the literature.
> - Section 6: We have clarified in the conclusion it will be important for future work to establish theoretical guarantees regarding the convergence rate of the GP and guarantees of convergence to LSPs.
> - Appendix, Table 2: We clarify that our reported success rates check the success of a given run against the true second-order optimality conditions of $f$ at the solution point.

---

> ### Author Response · Authors · 2024-11-18
> **Response by Authors [3/6]**
>
> **Comment:** There is no discussion on the rationale behind the assumptions in Lemma 4.1. Please offer an explanation here.
>
> **Response:** We thank the reviewer for the question. The assumptions of Lemma 4.1 are standard technical assumptions. Specifically, the main idea behind these assumptions is to ensure that the Jacobian of the players’ first order necessary conditions is nonsingular; this condition is well-known to ensure that Newton steps on those necessary conditions are descent directions on merit function $M^{CB}_t$ [Nocedal and Wright, Ch. 11], and ultimately that LLGame converges to a first-order local Nash point. If this point remains unclear, we are happy to update the manuscript.
>
> **Manuscript Changes:** We have clarified the rationale for these assumptions in Section 4.3.

---

> ### Author Response · Authors · 2024-11-18
> **Response by Authors [4/6]**
>
> **Comment:** The paper lacks a dedicated "Related Work" section. It appears that the related work has been split into two parts, placed separately in Section 1 and Section 2.  I suggest reorganizing Section 1 and Section 2. The current structure is somewhat disorganized and lacks a systematic introduction to prior work. The paper lacks a dedicated "Related Work" section. It appears that the related work has been split into two parts, placed separately in Section 1 and Section 2.
>
> **Response:** As requested, we have reorganized Sections 1 and 2 to more clearly introduce related work.
>
> **Manuscript Changes:** We renamed the second section of the manuscript to “Related Works and Preliminaries,” and moved the last two paragraphs of the introduction into this section to consolidate the related works and preliminaries into this section.

---

> ### Author Response · Authors · 2024-11-18
> **Response by Authors [5/6]**
>
> **Question:** How to incorporate second-order sufficient conditions to enhance the performance of your approach? Please give further explanation.
>
> **Response:** We believe the reviewer is likely referring to our statement about the possibility of incorporating second-order conditions to enhance the performance of our algorithms. We envision this arising within the LLGame problem, where currently we are only able to search for first order local Nash points. To understand why we do this, recall that the inner game is not zero-sum, but general-sum. Solving general-sum games in nonconvex-nonconcave settings with guarantees about satisfying second order conditions is an open problem. We note that prior works like Mazumdar et al. 2019 and Gupta 2024 et al. can solve zero-sum games in nonconvex-nonconcave settings with guarantees about second order conditions (though we note that for Mazumdar et al. 2019 and Gupta et al. 2024, such a guarantee is only provided if the algorithm converges). However, these approaches can not be applied to general-sum settings as in LLGame, and moreover they cannot be applied to the original zero-sum problem because they require access to second derivative information.
>
> Ultimately, then, our aim in this pointer to future work is to emphasize the importance of finding algorithms for finding (second order) local Nash equilibria in general-sum games, and to make it clear that progress in this direction would clearly benefit our framework.
>
> **Manuscript Changes:** We have made clarifications in Section 6 that more clearly explain how we aim to adjust our approach.

---

> ### Author Response · Authors · 2024-11-18
> **Response by Authors [6/6]**
>
> We hope these clarifications highlight the advantages of our work and make our contributions more transparent. We are more than happy to address any further concerns and provide additional explanations for any point. If we have addressed all your concerns, we kindly request that you reconsider our score.

---

> ### Comment · Reviewer_euUW · 2024-11-21
> **Reply**
>
> Thank you for your response! I will maintain my positive score and recommend accepting this paper.

---

> > ### Author Response · Authors · 2024-12-04
> >
> > As the review period comes to a close, we thank the reviewer once more for their questions and feedback, which have helped us improve our work significantly.
> > We would like to highlight a long discussion with reviewer gbJe which clarified a number of misunderstandings that the reviewer had regarding the placement of our work within the literature, the meaning and significance of the results on the zeroth-order, first-order, and second-order baselines we use, and additional results comparing to a related work. If the reviewer believes that this discussion may help them evaluate our work further, we invite them to take a closer look.
> >
> > Thank you once again for your time and effort.

---

### Official Review · Reviewer_C6CC · 2024-10-29

**Soundness:** 3
**Presentation:** 3
**Contribution:** 2
**Rating:** 5
**Confidence:** 3

**Summary:**

The paper studies the problem of finding local saddle point in two-player zero-sum games, in which the objective function is unknown and can only be accessed through zeroth-order samples. The authors approach the problem through Bayesian optimization, introducing a bi-level optimization algorithm where the lower-level solves a general-sum game and upper level uses Gaussian processes to build a surrogate model for the objective. The authors illustrate the algorithm performance through numerical simulations.

**Strengths:**

The paper targets an important problem, which is to solve two-player zero-sum games in the general nonconvex-nonconcave setting with only zeroth-order samples. The introduction and description the algorithm makes general sense. The numerical simulation verifies the algorithm performance to a certain extent.

**Weaknesses:**

My main concern is that the authors do not provide sufficient understandings for the proposed algorithm either in theory or through experiments.

The paper lacks rigorous mathematical justification, and the explanation on why the proposed algorithm is expected to work is made in a very hand-wavy manner. Most existing works on two-player zero-sum games come with end-to-end analysis on convergence rate/sample complexity. Rigorous theoretical results allow us to judge algorithms in terms of efficiency, restrictiveness of assumptions, etc. Without such theoretical results, it is hard to compare the proposed algorithm with those in the existing literature and those to be introduced in the future.

Regarding the takeaways highlighted in line 478-510 -- this is a plausible explanation that matches the observation from a few small-scale experiments. Much more extensive experiments need to be conducted to verify the statement. In the absence of theoretical convergence guarantees, systematic numerical simulations are especially important for understanding the algorithm performance, but they are currently missing.

**Questions:**

Minor comments:

1) The phrasing of the abstract and introduction line 043-046 gives the impression that the authors consider $f$ which is not necessarily differentiable. However, it turned out later a second-order differentiability assumption is needed. The abstract and introduction should be revised for better clarity.

2) In Algorithm 1 line 4 and Algorithm 2 line 7, you are not returning $\bar{x}^*,\bar{y}^*$/$x^*,y^*$. You are returning the iterates as an estimate of $\bar{x}^*,\bar{y}^*$/$x^*,y^*$.

---

> ### Author Response · Authors · 2024-11-18
> **Response by Authors [1/6]**
>
> We thank the reviewer for their detailed comments. Please find our responses below. All references refer to works cited in the manuscript.
>
> **Comment:** The paper targets an important problem, which is to solve two-player zero-sum games in the general nonconvex-nonconcave setting with only zeroth-order samples. The introduction and description of the algorithm makes general sense. The numerical simulation verifies the algorithm performance to a certain extent.
>
> **Response:** We thank the reviewer for their kind words regarding our contribution, and for their time and effort in reviewing our manuscript.

---

> > ### Author Response · Authors · 2024-11-18
> > **Response by Authors [2/6]**
> >
> > **Comment:** My main concern is that the authors do not provide sufficient understandings for the proposed algorithm either in theory or through experiments.
> >
> > The paper lacks rigorous mathematical justification, and the explanation on why the proposed algorithm is expected to work is made in a very hand-wavy manner. Most existing works on two-player zero-sum games come with end-to-end analysis on convergence rate/sample complexity. Rigorous theoretical results allow us to judge algorithms in terms of efficiency, restrictiveness of assumptions, etc. Without such theoretical results, it is hard to compare the proposed algorithm with those in the existing literature and those to be introduced in the future.
> >
> > **Response:** We thank the reviewer for their feedback on this important topic. We agree with the reviewer on the importance of establishing additional theoretical guarantees, but we believe that these empirical results are enough to establish the practical potential of the proposed black-box saddle point approach. As noted in our top-level comment, this work is intended as a practical first step in a new line of work towards solving black-box zero-sum games. Despite guaranteeing that LSS (a baseline method we take from the literature) finds a LSP when it converges, we note that the empirical performance of LSS demonstrates that it frequently does not succeed in converging on the challenging objectives, high-dimensional polynomial and decaying polynomials, accepted as common benchmark problems in the literature. Our algorithms generally outperform LSS (as shown in Appendix Table 2), finding LSPs at a similar rate at minimum but mostly outperforming it on these benchmark problems.
> >
> > In addition to Lemma 4.1, we highlight additional theoretical results Lemma A.1 and Remark A.2 which discuss how the LCB and UCB approach one another as more function evaluations are taken in subsequent iterations, meaning that the general-sum LLGame becomes more similar to the underlying zero-sum game as described in Section 4.
> >
> > **Manuscript Changes:**
> > - Section 2: Mention the drawbacks of existing methods
> > - Section 4.3: We more directly reference Lemma A.1 and Remark A.2 from the Appendix which discuss the convergence of LCB -and UCB under certain conditions.
> > - Section 5: We clarify that our chosen experiments are accepted as baseline problems by many previous works in the literature.
> > - Section 6: We have clarified in the conclusion how future work requires additional guarantees regarding convergence rate of the GP and guarantees of convergence to LSPs.
> > - Appendix, Table 2: We clarify that our reported success rates check the success of a given run against the true second-order optimality conditions of $f$ at the solution point.

---

> > > ### Author Response · Authors · 2024-11-18
> > > **Response by Authors [3/6]**
> > >
> > > **Comment:** Regarding the takeaways highlighted in line 478-510 -- this is a plausible explanation that matches the observation from a few small-scale experiments. Much more extensive experiments need to be conducted to verify the statement. In the absence of theoretical convergence guarantees, systematic numerical simulations are especially important for understanding the algorithm performance, but they are currently missing.
> > >
> > > **Response:** With respect, we believe that these experiments are extensive enough to establish the practical potential of the proposed black-box saddle point approach, though of course there remain a wide variety of interesting application problems which can and should be studied. We note that high-dimensional polynomial and the decaying polynomial examples are difficult problems taken from prior works [Bertsimas et al. 2010, Bogunovic et al. 2018, Mazumdar et al. 2019, and Gupta et al. 2024], and have been used in other papers as testbenches for comparing the performance of zero-sum game solvers. Moreover, as documented in Table 2 of the Appendix, our algorithms find LSPs more often than state-of-the-art baseline algorithms like LSS (because LSS only guarantees finding a LSP if it converges and it often fails to converge on these problems).
> > >
> > > Regarding the systematic numerical simulations, we refer the reviewer to Tables 1 and 2 of the Appendix, which contain additional details about convergence measured by number of Newton steps and success rate.
> > >
> > > We thank the reviewer for their time and effort, and we look forward to further suggestions which may improve our manuscript to clarify this topic.
> > >
> > > **Manuscript Changes:** We have included additional clarification in the manuscript regarding the results we have cited in our response. We clarify that our chosen experimental test cases are accepted as baseline problems by many previous works in the literature.

---

> > > > ### Author Response · Authors · 2024-11-18
> > > > **Response by Authors [4/6]**
> > > >
> > > > **Comment:** In Algorithm 1 line 4 and Algorithm 2 line 7, you are returning the iterates as an estimate of $\bar{x}^*, \bar{y}^* / x^*, y^*$.
> > > >
> > > > **Response:** We thank the reviewer for making note of this oversight. We have introduced language to clarify this nuance.
> > > >
> > > > **Manuscript Changes:**
> > > > - Section 2: We introduce language to specify that gradient-based algorithms run on the GP surrogate model return estimates of solutions.
> > > > - Section 4: We introduce language to remind the reader that optimizing objectives based on $\mu_t$ results in approximate solutions.

---

> > > > > ### Author Response · Authors · 2024-11-18
> > > > > **Response by Authors [5/6]**
> > > > >
> > > > > **Comment:** The phrasing of the abstract and introduction line 043-046 gives the impression that the authors consider $f$ which is not necessarily differentiable. However, it turned out later a second-order differentiability assumption is needed.
> > > > >
> > > > > **Response:** We thank the reviewer for catching this inconsistency in line 45. We had intended to refer to generic black box problems in this sentence, some of which do not assume differentiability. However, in order to discuss the definitions of local saddle point conditions, we have to assume $f$ is twice differentiable in our work. Nevertheless, we emphasize that our method does not require gradient access and can certainly be applied for nondifferentiable $f$; saddle point conditions can be at least approximately checked on the converged mean function of the GP, $\mu_t$.
> > > > >
> > > > > **Manuscript Changes:** In line 45, we remove the word “non-differentiable.”

---

> > > > > > ### Author Response · Authors · 2024-11-18
> > > > > > **Response by Authors [6/6]**
> > > > > >
> > > > > > We hope these clarifications highlight the advantages of our work and make our contributions more transparent. We are more than happy to address any further concerns and provide additional explanations for any point. If we have addressed all your concerns, we kindly request that you reconsider our score.

---

> > > > > > > ### Comment · Reviewer_C6CC · 2024-11-21
> > > > > > >
> > > > > > > I thank the authors for the detailed response. I agree that this paper is a first step in a new line of work towards solving black-box zero-sum games, a contribution that should be recognized. I do not think that there is something significant that I missed in the initial review phase, and thus would like to maintain my score.

---

> > > > > > > > ### Comment · Reviewer_gbJe · 2024-12-02
> > > > > > > >
> > > > > > > > Thanks for your responses.
> > > > > > > >
> > > > > > > > I have an issue here that I would appreciate your comments on. While I acknowledge the novelty of your method, I question the claim that it is the first algorithm for the zeroth-order black-box non-convex non-concave domain, given the existence of [1]. Although the *analysis* in [1] focuses on the convex-concave setting, this does not imply its algorithm is inapplicable to non-convex, non-concave scenarios, as most optimization methods for general settings are designed this way: inspired by convex optimization but applicable to general settings.
> > > > > > > >
> > > > > > > > In summary, in my opinion, your paper should either demonstrate that [1]'s algorithm is not applicable to non-convex settings or show that your algorithm performs comparably or better.
> > > > > > > >
> > > > > > > > [1] Zeroth-Order Methods for Convex-Concave Minmax Problems: Applications to Decision-Dependent Risk Minimization

---

> > > > > > > > > ### Author Response · Authors · 2024-12-04
> > > > > > > > > **Response to Questions About Claim [1/2]**
> > > > > > > > >
> > > > > > > > > **Comment:** Thanks for your responses.
> > > > > > > > >
> > > > > > > > > I have an issue here that I would appreciate your comments on. While I acknowledge the novelty of your method, I question the claim that it is the first algorithm for the zeroth-order black-box non-convex non-concave domain, given the existence of [1]. Although the analysis in [1] focuses on the convex-concave setting, this does not imply its algorithm is inapplicable to non-convex, non-concave scenarios, as most optimization methods for general settings are designed this way: inspired by convex optimization but applicable to general settings.
> > > > > > > > >
> > > > > > > > > In summary, in my opinion, your paper should either demonstrate that [1]'s algorithm is not applicable to non-convex settings or show that your algorithm performs comparably or better.
> > > > > > > > >
> > > > > > > > > [1] Zeroth-Order Methods for Convex-Concave Minmax Problems: Applications to Decision-Dependent Risk Minimization
> > > > > > > > >
> > > > > > > > > **Response:** We thank the reviewer for this insightful question and have carefully considered the question of whether the algorithm proposed by [1] is applicable in nonconvex-nonconcave settings. As the reviewer notes, the algorithms proposed in [1] can be run in these settings despite not being designed for them or tested on them. We implemented the SGDA-WR algorithm and ran experiments on the nonconvex-nonconcave decaying polynomial objective. Our results show that **SGDA-WR never converges on this testbench.** We summarize the success rates for each algorithm below.
> > > > > > > > >
> > > > > > > > > | Method           | Success Rate |
> > > > > > > > > |------------------|--------------|
> > > > > > > > > | Ef-Xplore (ours) | 60%          |
> > > > > > > > > | Ef-Xploit (ours)       | 30%          |
> > > > > > > > > | Exp-Xplore (ours)       | 60%          |
> > > > > > > > > | Exp-Xploit (ours)      | 50%          |
> > > > > > > > > | LSS              | 15%          |
> > > > > > > > > | SGDA-WR [1]         | 0%           |
> > > > > > > > >
> > > > > > > > > While we acknowledge that nothing in the implementation of the algorithm in [1] prevents it from being run on nonconvex-nonconcave settings, our experimental results show that it does not ever succeed on a standard problem in this setting. We note that a SGDA-WR regularly converged within 50 iterations on a convex-concave problem, and we reuse the same step size and query radius parameters that worked well on that problem. The paper provides a link to code, which is unfortunately invalid. Due to the short turnaround between the reviewer’s last comment and the end of the rebuttal period, we were not able to implement and validate the other algorithms from [1], but we expect the results to look similar and are willing to do this before a camera-ready version if requested.
> > > > > > > > >
> > > > > > > > > Based on this discussion and the reviewer’s feedback, we adjust our claim to the following. *To the best of our knowledge, this paper is the first to experimentally demonstrate an approach that (1) finds saddle points in black-box settings (2) on nonconvex-nonconcave objectives (3) with zeroth-order samples. Prior work either achieves only one or two of these simultaneously or fails to theoretically and experimentally validate the approach in nonconvex-nonconcave settings.*
> > > > > > > > >
> > > > > > > > > Lastly, we thank the reviewer for bringing up [1] for discussion and ensuring we were aware of its implications. We value its potential as a comparison to our method and have modified our manuscript to include it as a baseline and discuss it in further detail.
> > > > > > > > >
> > > > > > > > > **Manuscript Changes (not uploaded yet):**
> > > > > > > > > - Section 1/2: Discussed more about [1] specifically and its position in the literature, including nuances about how it can run in this setting.
> > > > > > > > > - Section 5: Added additional introduction to this baseline.
> > > > > > > > > - Figure 1, 4 OR Table 2: Added [1] as a baseline reported in the figures.
> > > > > > > > > - Appendix: Described the way we implemented this algorithm (hyperparameters and experimental setup).
> > > > > > > > > As the rebuttal period is closing, we have not been able to upload our manuscript. However, the promised changes will be introduced in a future draft.

---

> > > > > > > > > > ### Author Response · Authors · 2024-12-04
> > > > > > > > > > **Response to Questions About Claim [2/2]**
> > > > > > > > > >
> > > > > > > > > > We thank the reviewer once more for their diligence in reviewing our work. We sincerely enjoyed our discussion, and our manuscript has improved significantly.

---

> > > > > > > > ### Author Response · Authors · 2024-12-04
> > > > > > > >
> > > > > > > > As the review period comes to a close, we thank the reviewer once more for their questions and feedback, which have helped us improve our work significantly.
> > > > > > > > We would like to highlight a long discussion with reviewer gbJe which clarified a number of misunderstandings that the reviewer had regarding the placement of our work within the literature, the meaning and significance of the results on the zeroth-order, first-order, and second-order baselines we use, and additional results comparing to a related work. If the reviewer believes that this discussion may help them evaluate our work further, we invite them to take a closer look.
> > > > > > > >
> > > > > > > > Thank you once again for your time and effort.

---

### Official Review · Reviewer_gbJe · 2024-11-04

**Soundness:** 3
**Presentation:** 3
**Contribution:** 3
**Rating:** 6
**Confidence:** 3

**Summary:**

This paper proposes a new framework for finding local saddle points in zeroth-order black-box saddle point optimization.
The authors propose a bilevel framework inspired by Bayesian Optimization (BO) that leverages Gaussian Processes (GPs) to model the unknown objective function. The lower-level game uses confidence bounds from the GP to define a general-sum game, the solution of which (a local Nash point) is used to guide sampling of the objective function. The high-level game updates the GP model with the new samples and iteratively searches for a local saddle point. The empirical results show that BSP variants outperform baseline approaches like naive random sampling and zeroth-order gradient descent-ascent on these problems.

**Strengths:**

This paper addresses a problem that has received limited attention in the literature: the optimization of black-box games. It introduces a novel framework that combines a diverse set of techniques to tackle this challenge, making it a noteworthy contribution.

**Weaknesses:**

- Limitations of modeling with GP: We know that the required number of samples to accurately model the objective function grows exponentially, leading to increased computational complexity and potentially reduced accuracy. Do you plan to address this limitation in future work?
- The experiments are ultimately not convincing: while the experiments on synthetic data successfully achieve the optimal merit function, the real-world examples are less successful compared to the only other baseline and require further improvement.
- Lack of sufficient theoretical guarantees for the method: While Lemma 4.1 demonstrates the convergence of LLGAME to a Local Nash Point (LNP), the paper lacks a theoretical analysis proving the convergence of the overall Bayesian Saddle Point (BSP) algorithm to a Local Saddle Point (LSP). This is particularly important, as the authors acknowledge that the LNP returned by LLGAME might not necessarily be an LSP of the original objective function.

**Questions:**

Did you consider comparing your method with first-order algorithms? Incorporating them as a baseline can provide valuable insights.

---

> ### Author Response · Authors · 2024-11-18
> **Response by Authors [1/6]**
>
> We thank the reviewer for their detailed comments. Please find our responses below. All references refer to works cited in the manuscript.
>
> **Comment:** This paper addresses a problem that has received limited attention in the literature: the optimization of black-box games. It introduces a novel framework that combines a diverse set of techniques to tackle this challenge, making it a noteworthy contribution.
>
> **Response:** We thank the reviewer for their kind comments on the strengths of our work, and for their time and effort in reviewing our manuscript.

---

> > ### Author Response · Authors · 2024-11-18
> > **Response by Authors [2/6]**
> >
> > **Comment:** Lack of sufficient theoretical guarantees for the method: While Lemma 4.1 demonstrates the convergence of LLGAME to a Local Nash Point (LNP), the paper lacks a theoretical analysis proving the convergence of the overall Bayesian Saddle Point (BSP) algorithm to a Local Saddle Point (LSP). This is particularly important, as the authors acknowledge that the LNP returned by LLGAME might not necessarily be an LSP of the original objective function.
> >
> > **Response:** We thank the reviewer for their feedback on this important topic. As noted in our top-level comment, this manuscript opens up a new line of work in solving black-box zero-sum games. While we acknowledge that our algorithms may converge to a point which is not precisely a LNP, our empirical success rate results shown in Appendix Table 2 demonstrate that our algorithms find LSPs more frequently than a state-of-the-art baseline method from the literature (called LSS) even when it is provided with a gradient oracle. In the case of LSS, this is because the theoretical guarantee that LSS’ limit points coincide with local saddle points only holds if LSS converges. In practice, we find that LSS often fails to converge on common experiments used in the literature, as reported in Appendix Table 2. Thus, we believe that these empirical results are enough to establish the practical potential of the proposed black-box saddle point approach.
> >
> > In addition to Lemma 4.1, we highlight additional theoretical results Lemma A.1 and Remark A.2 which discuss how the LCB and UCB approach one another as more function evaluations are taken in subsequent iterations, meaning that the general-sum LLGame becomes more similar to the underlying zero-sum game as described in Section 4.
> >
> > **Manuscript Changes:**
> > - Section 2: We provide a clearer discussion of the drawbacks of existing methods.
> > - Section 4.3: We more directly reference Lemma A.1 and Remark A.2 from the Appendix which discuss the convergence of LCB and UCB under certain conditions.
> > - Section 5: We clarify that our chosen experiments are drawn from example problems in the literature.
> > - Section 6: We have clarified in the conclusion it will be important for future work to establish theoretical guarantees regarding the convergence rate of the GP and guarantees of convergence to LSPs.
> > - Appendix, Table 2: We clarify that our reported success rates check the success of a given run against the true second-order optimality conditions of $f$ at the solution point.

---

> ### Author Response · Authors · 2024-11-18
> **Response by Authors [3/6]**
>
> **Comment:** The experiments are ultimately not convincing: while the experiments on synthetic data successfully achieve the optimal merit function, the real-world examples are less successful compared to the only other baseline and require further improvement.
>
> **Response:** We thank the reviewer for this feedback. With respect, we believe that these experiments are extensive enough to establish the practical potential of the proposed black-box saddle point approach, though of course there remain a wide variety of interesting application problems which can and should be studied. Regarding the “real-world examples,” we assume that the reviewer is referring to the ARIMA MPC example from Figure 2. In this particular example (see the key takeaway of Figure 2), we highlight the 27.6% improvement in the out-of-distribution (OoD) cost when using our version of Robust MPC and have revised the language to emphasize why operating at a saddle point should lead to better performance on this metric, compared with a MPC which is not operating at a saddle point. Additionally, the third column of Figure 4 in the Appendix demonstrates that all of our algorithms achieve the optimal merit function value when provided a large number of initial samples on this real-world example, and that three of the four do so when provided a limited number of initial samples.
>
> We note that high-dimensional polynomial and the decaying polynomial examples are difficult problems taken from prior works [Bertsimas et al. 2010, Bogunovic et al. 2018, Mazumdar et al. 2019, and Gupta et al. 2024], and that they have been used in other papers as testbenches for comparing the performance of zero-sum game solvers. Moreover, as documented in Table 2 of the Appendix, our algorithms find LSPs more often than state-of-the-art baseline algorithms like LSS (because LSS only guarantees finding a LSP if it converges).
>
> **Manuscript Changes:** We have included additional clarification in the manuscript regarding the results we have cited in our response. We clarify that our chosen experimental test cases are accepted as baseline problems by many previous works in the literature.
>
> We thank the reviewers again for their time and effort and look forward to further suggestions which may improve our manuscript to clarify this topic.

---

> > ### Author Response · Authors · 2024-11-18
> > **Response by Authors [4/6]**
> >
> > **Comment:** Limitations of modeling with GP: We know that the required number of samples to accurately model the objective function grows exponentially, leading to increased computational complexity and potentially reduced accuracy. Do you plan to address this limitation in future work?
> >
> > **Response:** We thank the reviewer for pointing out this drawback of using GPs. We agree with the reviewer that adapting methods to handle higher dimensions is a possible avenue of future work. One possible method by which we can explore it is by improving sampling methods for the GP given knowledge of where LLGame may converge.
> >
> > **Manuscript Changes:** In Section 2, we introduce language in the preliminaries to remind the reader of these drawbacks. In Section 6, we adjust some details about future work.

---

> > > ### Author Response · Authors · 2024-11-18
> > > **Response by Authors [5/6]**
> > >
> > > **Question:** Did you consider comparing your method with first-order algorithms? Incorporating them as a baseline can provide valuable insights.
> > >
> > > **Response:** We thank the reviewer for this insightful question. To our knowledge, no existing work directly addresses the problem of black-box saddle point optimization with zeroth-order samples. Thus, we chose the random and zeroth-order baselines as naive methods to compare against. Additionally, we considered modifying existing methods from [Adolphs et al. 2019, Mazumdar et al. 2019, Gupta et al. 2024]. Of these three, only Mazumdar et al. 2019 (which introduces LSS) handles noisy samples though it requires first- and second-order gradient samples. As our setting involves only zeroth-order samples, we provide LSS a noisy oracle which provides access to the first- and second-order gradients. We considered two other options to adapt LSS to the zeroth-order sample setting: first, finite differencing zeroth-order samples to compute first- and second-order derivatives. Second, we considered an oracle which provides first-order derivative samples and requires LSS to use finite differencing for the second-order derivatives. However, the results from both of these methods were similar to the zeroth-order baseline in that finite differencing was sample-inefficient and resulted in excessive noise. Thus, we provided LSS with an oracle providing the most information, first- and second-order gradient samples, to provide the most useful comparison to our method.

---

> > > > ### Author Response · Authors · 2024-11-18
> > > > **Response by Authors [6/6]**
> > > >
> > > > We hope these clarifications highlight the advantages of our work and make our contributions more transparent. We are more than happy to address any further concerns and provide additional explanations for any point. If we have addressed all your concerns, we kindly request that you reconsider our score.

---

> > > > ### Comment · Reviewer_gbJe · 2024-11-22
> > > > **Choice of baselines;**
> > > >
> > > > The choice of LLS, a second-order method, as a baseline is unclear. Why not use a simpler alternative like gradient descent ascent with finite differences? Moreover, back to my original point, in your polynomial example, where gradients are accessible, it would be highly informative to compare your method against a first-order approach, such as gradient descent ascent. As expected, a zeroth-order method would likely perform worse, but such a comparison would provide valuable context for evaluating your approach.
> > > >
> > > > Additionally,  I believe an important reference by Maheshwari et al. may have been overlooked:  [Zeroth-Order Methods for Convex-Concave Min-max Problems: Applications to Decision-Dependent Risk Minimization](https://proceedings.mlr.press/v151/maheshwari22a/maheshwari22a.pdf).
> > > > This work introduces a zeroth-order method for convex-concave problems, which could also potentially serve as a baseline for your experiments. Citing it as related work is crucial to properly position your contribution within the existing literature.
> > > >
> > > > I’ve reviewed the updated version of your paper and read through your detailed responses. Thank you for providing them. After consideration, I believe it’s best to maintain my original score.

---

> ### Author Response · Authors · 2024-11-23
> **Response by Authors to Choice of Baselines [1/4]**
>
> **Important Clarification:** We thank the reviewer for their comments. Prior to responding to the specific comments, we would like to clarify a point regarding our contributions compared to prior work, which will be critical to our subsequent responses.
> Prior work on saddle point optimization varies across three axes:
> - A1. whether the optimization objective is known or sampled (“black-box”),
> - A2. whether the optimization objective is convex-concave or nonconvex-nonconcave, and
> - A3. whether the method assumes gradient access for the samples or not.
>
> We present the first black-box technique for saddle point optimization on nonconvex-nonconcave objectives based on zeroth-order information. While prior works may exist that find saddle points in black-box settings or on nonconvex-nonconcave objectives or with zeroth-order samples, our work is the first, to our knowledge, that achieves all three simultaneously.
>
> **Manuscript Changes:** Upon reviewing the manuscript once more, we realize we did not state this sufficiently clearly within our contribution, and we have updated the manuscript to clarify this. Specifically, we did not note the nonconvexity-nonconcavity of the objective in our contributions at all, and wanted to clarify in case this had caused confusion.

---

> ### Author Response · Authors · 2024-11-23
> **Response by Authors to Choice of Baselines [2/4]**
>
> **Comment:** The choice of LLS, a second-order method, as a baseline is unclear. Why not use a simpler alternative like gradient descent ascent with finite differences? Moreover, back to my original point, in your polynomial example, where gradients are accessible, it would be highly informative to compare your method against a first-order approach, such as gradient descent ascent. As expected, a zeroth-order method would likely perform worse, but such a comparison would provide valuable context for evaluating your approach.
>
> We thank the reviewer for this line of questioning, as we may have misunderstood the previous question. As mentioned in the manuscript, we provide three baselines:
> - Random: This baseline is a zeroth-order optimization method which randomly samples points in the domain and keeps the one with the lower cost. We note, for the purpose of clarity, that zeroth-order here refers to both the optimization method (which does not require gradients at all as compared to GDA) and the order of the samples (each of which simply includes the noisy cost measurement).
> - Zeroth: This baseline refers to gradient descent-ascent (GDA) with finite differencing to compute the gradient samples. While we do note this in Section 5.1, in retrospect, this is inappropriately named, and we have changed the name to “GDA with FD”. We also note that the merit function for GDA with FD never converges to 0 (see Figure 1) on our example problems due to the noise introduced by finite differencing, which is excessive for the problems we test against, and subsequently prevents convergence on those problems.
> - Local Symplectic Surgery (LSS) [Mazumdar et al. 2019]: Returning to the reviewer's question about LSS, we sought to compare our work to a state-of-the-art approach in order to compare our work with others which have additional theoretical guarantees on these problems. We were limited to works in nonconvex-conconcave settings (A2), and only Mazumdar et al. 2019 provided convergence guarantees in a black-box setting (A1). Thus, we chose to include LSS as a state-of-the-art baseline despite it requiring gradient access (A3). Due to finite differencing not working well in the GDA baseline, we chose to provide a gradient oracle instead of finite differencing, as noted in our previous comment. We note that when we did try LSS  on the decaying polynomial example problem with finite differencing to estimate first- and second-order gradients, the solver always converged to the false saddle point at (0, 0).
>
> With regards to using GDA without finite differencing, if this is what the reviewer meant, we don’t quite understand the benefit over the first-order method we did include, but we are open to hearing more.
>
> In summary, we provide zeroth-order, first-order, and second-order baselines as comparisons to our work, including the exact baseline the reviewer asked about (albeit badly labeled). We hope this response clarifies our decision to use each baseline.
>
> **Manuscript Changes:** Renamed “Zeroth” baseline to GDA with FD in Figure 1 and Section 5. We also added additional discussion about baselines in Section 5.

---

> > ### Author Response · Authors · 2024-11-23
> > **Response by Authors to Choice of Baselines [3/4]**
> >
> > **Comment:** Additionally, I believe an important reference by Maheshwari et al. may have been overlooked: Zeroth-Order Methods for Convex-Concave Min-max Problems: Applications to Decision-Dependent Risk Minimization. This work introduces a zeroth-order method for convex-concave problems, which could also potentially serve as a baseline for your experiments. Citing it as related work is crucial to properly position your contribution within the existing literature.
> >
> > **Response:** We thank the reviewer for bringing our attention to this work. We acknowledge that our related works section currently lacks prior works that assume zeroth-order information. Upon closer review of this particular work, we note that it involves black-box minimax optimization on convex-concave objectives with zeroth-order information: hence, it differs from our work on axis A2 (convexity-concavity) though it involves black-box optimization (axis A1) without gradient information (axis A3) as with our work. Thus, it may not be appropriate to use as a baseline. Nevertheless, we acknowledge its importance as a crucial related work to cite.
> >
> > **Manuscript Changes:** We introduce this citation in our related work as discussion related to zeroth-order methods.

---

> > > ### Author Response · Authors · 2024-11-23
> > > **Response by Authors to Choice of Baselines [4/4]**
> > >
> > > **Comment:** I’ve reviewed the updated version of your paper and read through your detailed responses. Thank you for providing them. After consideration, I believe it’s best to maintain my original score.
> > >
> > > **Response:** Thank you for your detailed response. This review exposed some unclear areas in our work, and the further clarification has improved our work greatly. If the reviewer has further comments or questions, we would love to hear the feedback.

---

> > ### Comment · Reviewer_gbJe · 2024-11-25
> > **Reply to author's response**
> >
> > Thank you for your response.
> >
> > The explanation regarding your random and zeroth-order baselines, as well as the new naming, was very helpful. I appreciate it.
> >
> > The fact that you are providing the first- and second-order oracles to LSS (if I understand correctly) and yet your method still performs better highlights the strength of your approach.
> >
> > To ensure the accuracy of this result, could you clarify what the merit function is exactly? I am surprised that I could not find an exact equation for it. Describing your only metric vaguely with words is not very helpful. This is what I found: *$M_f$, which is calculated based on the true gradients of the underlying function, rather than the confidence bounds employed in the actual algorithm.* Based on this description, and assuming you have access to the function for the first two experiments, I would expect you to use the second-order derivative to ensure the algorithms do not converge to spurious saddle points. Is that what you did?

---

> ### Author Response · Authors · 2024-11-26
> **Response by Authors About the Merit Function [1/2]**
>
> **Comment:** Thank you for your response.
>
> The explanation regarding your random and zeroth-order baselines, as well as the new naming, was very helpful. I appreciate it.
>
> The fact that you are providing the first- and second-order oracles to LSS (if I understand correctly) and yet your method still performs better highlights the strength of your approach.
>
> **Response:** We thank the reviewer for their kind words, and for helping us clarify this point to better highlight the strengths of our approach. We confirm that the reviewer’s understanding is correct. Our method performs better than LSS in this setting despite us providing LSS with an “unfair advantage,” so to speak: providing it first- and second-order oracles.

---

> ### Author Response · Authors · 2024-11-26
> **Response by Authors About the Merit Function [2/2]**
>
> **Comment:** To ensure the accuracy of this result, could you clarify what the merit function is exactly? I am surprised that I could not find an exact equation for it. Describing your only metric vaguely with words is not very helpful. This is what I found: $M_f$, which is calculated based on the true gradients of the underlying function, rather than the confidence bounds employed in the actual algorithm. Based on this description, and assuming you have access to the function for the first two experiments, I would expect you to use the second-order derivative to ensure the algorithms do not converge to spurious saddle points. Is that what you did?
>
> **Response:** We thank the reviewer for this question about the merit function, and for the opportunity to clarify.
>
> For the definition of merit function $M^f$, we point the reviewer to Equation (8) in Section 4.2, where we define $M^f$ as follows,
>
> \\[G^{f}(x, y) = \left[\begin{array}{c} \nabla_{x} f (x, y)  \\ -\nabla_{y} f (x, y) \end{array}\right]^\intercal, ~~~~~ M^{f}(x, y) = \frac{1}{2} \|\|G^{f}(x, y) \|\|^2_2. \\]
>
> Thus, we note that minimizing $M^f$ satisfies first-order necessary conditions as defined in Proposition 3.3. This is the merit function used to produce the plots in Figures 1 and 4.
>
> As the reviewer correctly expects, we then compute the success rate based on the true second-order derivatives $\nabla^2_{xx} f, \nabla^2_{yy} f$ as described by the second-order sufficient conditions in Proposition 3.4. This ensures that we do not include spurious saddle points when we report success rates in Appendix Table 2.
>
> Next, for the purpose of clarity, we reiterate the differences between the three merit functions present in our manuscript, $M^{f}$, $M\^{CB}\_{t}$, and $M\^{\\mu}\_{t}$. Each of these is associated with a vector $G^{f}$, $G\^{CB}\_{t}$, and $G\^{\\mu}\_{t}$, which describe the roots which satisfy first-order conditions according to Proposition 3.3. For the latter two, the subscript $t$ indexes the iterations of BSP, i.e. how many times we have sampled the function, added the sample to the dataset, and updated the GP estimate of $f$.
>
> As $f$ is not known in our black-box setting, we can not use it directly to measure progress within the algorithm. Thus, we do the following in our work.
> - $M^f$ measures the progress towards the true first-order roots of the true objective, and it is used solely for describing the performance of our algorithm compared to other algorithms. It is **not** used within the BSP algorithm at all.
> - $G^{CB}_t$ is defined using the gradients of $\\text{LCB}_t$ with respect to x and $-\text{UCB}_t$ with respect to y. The confidence bounds are defined as a function of the GP estimate. Thus, $M^{CB}_t$ measures the progress towards a first-order root of the general-sum lower-level game (5). $M^{CB}_t$ is only used in the lower level game.
> - $M\^{\\mu}\_{t}$ serves as the replacement of $M^{f}$ in our BSP algorithm. The merit function $M\^{\\mu}\_{t}$ is defined by replacing $f$ in $M^{f}$ with $\\mu_t$, the mean function of the GP at iteration $t$. Thus, $M\^{\\mu}\_{t}$ measures the progress of the high-level, zero-sum game towards a first-order root of the GP mean $\\mu_t$, which estimates $f$ based on the samples. Upon the convergence of the algorithm, BSP tests the second-order conditions (in Proposition 3.4) using $\\nabla_{xx} \\mu_t$ and $\\nabla_{yy} \\mu_t$ to test for spurious saddle points. If the test fails, it reinitializes and begins again.
>
> We hope that this answers the reviewer’s question, but if anything remains unclear, we are happy to further refine our manuscript based on feedback.
>
> **Manuscript Changes:** We edit the passage in Section 5 that the reviewer cites to include a reference to Eq. (8) in Section 4.2, where $M^f$ is defined. We further clarify in the caption of Table 2 that the success rate is checked using the true second-order sufficient conditions of $f$, as defined in Proposition 3.4.

---

> > ### Author Response · Authors · 2024-12-02
> >
> > We thank the reviewer for their time and effort in engaging with us and providing valuable feedback to improve our work. If there are any further questions or concerns regarding our work as the rebuttal period comes to a close, we are more than happy to provide further clarifications.

---

### Author Response · Authors · 2024-11-18
**Top-Level Comment by Authors - Thank you for your time and effort to help us improve our work!**

We thank the reviewers for their kind words about our work and their time and effort in helping us improve our manuscript with their insightful questions and comments.

We would like to highlight to the reviewers that the primary goal of our work is to recognize an important open problem in the current literature: saddle point optimization in the black-box setting on nonconvex-nonconcave objectives with zeroth-order samples. Our findings show that existing methods are not applicable to this problem, and we propose an initial approach aimed at motivating further research. Our approach demonstrates several key design principles—such as lower- and higher-level game structures—that we believe will be critical for future studies, alongside strong experimental results. We hope that our work inspires further research in these directions, and in particular we acknowledge the importance of establishing theoretical grounding for these design principles. *To the best of our knowledge, this paper is the first to experimentally demonstrate an approach that (1) finds saddle points in black-box settings (2) on nonconvex-nonconcave objectives (3) with zeroth-order samples. Prior work either achieves only one or two of these simultaneously or fails to theoretically and experimentally validate the approach in nonconvex-nonconcave settings.*

We respond directly to reviewers’ comments below. For the purpose of clarifying our changes, we have uploaded an updated version of our manuscript with changes highlighted in pink. A second version responding to the reviewers' second round of comments highlights new changes in orange. We will remove this highlighting in a further update before the review period ends. We look forward to additional feedback, and are happy to make further changes to clarify and improve our work.

Update on November 27, 2024: We have removed the highlighted portions of the text, as there is a deadline that prevents further draft edits after today. To easily identify further changes, we refer reviewers to the "Manuscripts Changes" section of each comment where we highlight changes based on the reviewer's comments.

---

> ### Author Response · Authors · 2024-12-04
>
> We thank the reviewers one final time for their feedback and engagement during the discussion period. We are happy the reviewers like that our approach to the relevant and understudied problem of saddle point optimization used a diverse set of techniques based on Bayesian optimization to create a surrogate model of the objective (gbJe, C6CC, euUW), introduced a way to frame the saddle point optimization as a bilevel game (gbJe, C6CC, euUW), and strong, thorough experimental results which outperformed existing baselines (gbJe, eeUW).
>
> We have addressed each reviewer’s comments thoroughly, including by
> - **reiterating** the diversity, strength, and thoroughness of our **first-of-kind experimental results on multiple challenging real-world and synthetic settings,** (as emphasized in the previous comment)
> - **introducing a new baseline and associated experimental results** based on a related work we were initially unaware of, **which our method outperforms significantly** on nonconvex-nonconcave settings, and
> - clarifying the reasoning behind each choice of **baseline, including zeroth-order, first-order and second-order comparisons.**

---

### Meta-Review · Area_Chair_oS5g · 2024-12-21

**Metareview:**

This paper proposes a method to find local saddle points in black box two-player zero-sum games with potentially non-convex non-concave objective functions.

The strength of this work is that it provides an algorithm to tackle a challenging problem (namely,  black box two-player zero-sum games with potentially non-convex non-concave objective functions).

The main two weaknesses of this work are:
- The authors do not provide novel theoretical guarantees (Lemma 4.1 is an application of a standard theorem on Newton's method and only regards the inner loop). Even though it has been acknowledged by the authors, it remains a weakness. I believe it would be interesting to get guarantees in a (theoretically) tractable setting such as convex-concave.
- The experiments only deal with toyish low-dimensional problems.
If we reflect on the motivation for this setting provided by the authors:
> For example, in robust portfolio optimization, the goal is to create portfolios resistant to stock market fluctuations (Nyikosa, 2018), which are inherently random and difficult to model but can be sampled in a black-box fashion through trial and error. Similar problems arise in various physical settings, such as robotics (Lizotte et al., 2007) and communication networks (Qureshi & Khan, 2023)

The experiments provided by the authors do not reflect the ' Motivat[ing] ... real-world examples'  mentioned in the introduction and thus undermine one of the main claim of this work, which is that 'We experimentally demonstrate our algorithms’ effectiveness on a variety of challenging synthetic and realistic datasets.'

In particular, regarding the ARIMA application, the space is 4 dimensional, the noise is Gaussian (thus, it remains a synthetic experiment), and the baseline with random search is not compared.

**Additional Comments On Reviewer Discussion:**

The reviewers all noted the weaknesses mentioned in the rebuttal. (except  Reviewer euUW who did not mention the experiments).

Reviewer gbJe increased their score without providing any justification, even after I asked for some more feedback. I assume that it resulted from the discussion with the author, which I read in detail. Note that they originally mentioned that they would 'maintain their score'. I believe that most of the weaknesses originally mentioned by the reviewers still remain and are accounted for in my final recommendation.

---

### Decision · Program_Chairs · 2025-01-22

Reject